# PacGAN: The power of two samples in generative adversarial networks

**Zinan Lin**
ECE Department
Carnegie Mellon University
zinanl@andrew.cmu.edu

**Ashish Khetan**
IESE Department
University of Illinois at Urbana-Champaign
ashish.khetan09@gmail.com

**Giulia Fanti**
ECE Department
Carnegie Mellon University
gfanti@andrew.cmu.edu

**Sewoong Oh**
IESE Department
University of Illinois at Urbana-Champaign
swoh@illinois.edu

## Abstract

Generative adversarial networks (GANs) are a technique for learning generative models of complex data distributions from samples. Despite remarkable advances in generating realistic images, a major shortcoming of GANs is the fact that they tend to produce samples with little diversity, even when trained on diverse datasets. This phenomenon, known as mode collapse, has been the focus of much recent work. We study a principled approach to handling mode collapse, which we call *packing*. The main idea is to modify the discriminator to make decisions based on multiple samples from the same class, either real or artificially generated. We draw analysis tools from binary hypothesis testing—in particular the seminal result of Blackwell [4]—to prove a fundamental connection between packing and mode collapse. We show that packing naturally penalizes generators with mode collapse, thereby favoring generator distributions with less mode collapse during the training process. Numerical experiments on benchmark datasets suggest that packing provides significant improvements.

## 1 Introduction

Generative adversarial networks (GANs) are a technique for training generative models to produce realistic examples from an unknown data distribution [10]. Suppose we are given $N$ i.i.d. samples $X_1, \ldots, X_N$ from an unknown probability distribution $P$ over some high-dimensional space $\mathbb{R}^p$ (e.g., images). The goal of generative modeling is to learn a model that can draw samples from distribution $P$. In data-driven generative modeling, this model is typically formulated as a function $G : \mathbb{R}^d \to \mathbb{R}^p$ that maps a low-dimensional *code vector* $Z \in \mathbb{R}^d$ drawn from a standard distribution (e.g. spherical Gaussian) to a high-dimensional domain of interest.

A breakthrough in training such generative models was achieved by the innovative idea of GANs. GANs train two neural networks called the *generator* $G(Z)$ and *discriminator* $D(X)$. The role of the generator is to produce realistic samples, and the role of the discriminator is to distinguish generated samples from real data. These two neural networks play a dynamic minimax game against each other. If trained long enough, eventually the generator learns to produce samples that are indistinguishable from real data (but preferably different from the training samples). Concretely, GANs search for the

parameters of neural networks $G$ and $D$ that optimize the following minimax objective:

$$
\begin{aligned}
G^* \quad \in \quad & \arg\min_{G} \ \max_{D} \ V(G, D) \\
= \quad & \arg\min_{G} \ \max_{D} \ \mathbb{E}_{X \sim P}[\log(D(X))] + \\
& \mathbb{E}_{Z \sim P_Z}[\log(1 - D(G(Z)))] \, ,
\end{aligned}
\tag{1}
$$

where $P$ is the distribution of the real data, and $P_Z$ is the distribution of the input code vector $Z$. Critically, [10] shows that the global optimum of (1) is achieved if and only if $P = Q$, where $Q$ is the generated distribution of $G(Z)$. The solution to the minimax problem (1) can be approximated by iteratively training two "competing" neural networks, the generator $G$ and discriminator $D$. Each model can be updated by backpropagating the gradient of the loss function to its parameters.

A major challenge in training GANs is a phenomenon known as *mode collapse*, which refers to a lack of diversity in generated samples. Indeed, GANs commonly miss modes when trained on multimodal distributions. For instance, when trained on hand-written digits with ten modes, the generator might fail to produce some of the digits [24]. Several approaches have been proposed to fight mode collapse, e.g. [7, 8]. Proposed solutions rely on modified architectures, loss functions, and optimization algorithms. Although each of these proposed methods empirically mitigates mode collapse, we lack rigorous explanations of why the empirical gains are achieved—especially when those gains are sensitive to hyperparameters.

**Our Contributions.** In this work, we examine GANs through the lens of *hypothesis testing*. By viewing the discriminator as performing a binary hypothesis test on samples (i.e., whether they were drawn from distribution $P$ or $Q$), we can apply classical hypothesis testing results to the analysis of GANs. This view leads to three contributions:

(1) *Conceptual*: We propose a formal definition of mode collapse that abstracts away the geometric properties of the underlying data distributions (Section 3). This definition is closely related to the notion of ROC curves in binary hypothesis testing. Given this definition, we provide a new interpretation of the pair of distributions $(P, Q)$ as a two-dimensional region called the *mode collapse region*, where $P$ is the true data distribution and $Q$ the generated one. The mode collapse region provides new insights on how to reason about the relationship between those two distributions.

(2) *Analytical*: Through the lens of hypothesis testing and mode collapse regions, we show that if the discriminator is allowed to see samples from the $m$-th order product distributions $P^m$ and $Q^m$ instead of the usual target distribution $P$ and generator distribution $Q$, then the corresponding loss when training the generator naturally penalizes generator distributions with strong mode collapse (Section 3). Hence, a generator trained with this type of discriminator will choose distributions that exhibit less mode collapse. The *region* interpretation of mode collapse and corresponding data processing inequalities provide novel analysis tools for proving strong and sharp results with simple proofs. Technically, this leads to a novel *geometric* analysis technique to find the optimal solutions of infinite dimensional non-convex optimization problems of interest in Eqs. (2) and (3).

(3) *Algorithmic*: We propose a new GAN framework to mitigate mode collapse, which we call PacGAN. PacGAN can be seamlessly applied to existing GANs, requiring only a small modification to the discriminator architecture (Section 2). The key idea is to pass $m$ "packed" or concatenated samples to the discriminator, which are jointly classified as either real or generated. This allows the discriminator to do binary hypothesis testing based on the product distributions $(P^m, Q^m)$, which naturally penalizes mode collapse (Section 3). We demonstrate on benchmark datasets that PacGAN significantly improves upon competing approaches in mitigating mode collapse (Section 4), notably *minibatch discrimination* [24].

**Related Work** Three primary challenges appear in the GAN literature: ($i$) they are unstable to train, ($ii$) they are challenging to evaluate, and ($iii$) they exhibit mode collapse (more broadly, they do not generalize). Our work explicitly addresses challenge ($iii$), which is the focus of this section.

Mode collapse is a byproduct of poor generalization—i.e., the generator does not learn the true data distribution; this phenomenon is of significant interest [2, 3, 18, 1, 2]. Prior work has observed two types of mode collapse: entire modes from the input data are never generated, or the generator only creates images within a subset of a particular mode [9, 27, 3, 7, 20, 23]. These phenomena are not well-understood, but a number of explanatory hypotheses have been proposed, including improper objective functions [1, 2] and weak discriminators [20, 24, 2, 17]. Building on the second hypothesis,

we show that a packed discriminator can significantly reduce mode collapse, both theoretically and in practice. We compare packing to three main approaches for mitigating mode collapse:

(1) *Joint Architectures.* In encoder-decoder architectures, the GAN learns an encoding $G^{-1}(X)$ from the data space to a lower-dimensional latent space, in addition to the usual decoding $G(Z)$ from the latent space to the data space (e.g., BiGAN [8], adversarially learned inference [7], VEEGAN [25]). Despite empirical gains in such joint architectures, we find that packing captures more modes for a fixed generator and discriminator architecture, with less architectural and computational overhead. Also, recent work suggests that such architectures may be unable to prevent mode collapse [2].

(2) *Augmented Discriminators.* Several proposals have strengthened the discriminator by providing it with image labels [5] and/or more samples. A latter approach, *minibatch discrimination* [24], feeds an array of data samples to the discriminator, which uses the minibatch as side information to classify each sample individually. Recent work improved minibatch discrimination by progressively training discriminators on larger minibatches, with impressive visual results [13]. While packing and minibatch discrimination start from the same intuition that showing multiple samples at the discriminator helps mitigate mode collapse, how this idea is implemented in the discriminator architectures are completely different. PacGAN is easier to implement, empirically effective, and our theoretical analysis shows that packing is a principled way to use batched samples. For example, in the experiment in Appendix B.2 (left column of Table 6), the default DCGAN discriminator has $585$ weights in total in the Unrolled GAN implementation, the proposed PacDCGAN4 only adds 162 more weights to the discriminator, while minibatch discriminator adds 1,225,732 more weights.

(3) *Optimization-based solutions.* GANs are typically trained with iterative generator-discriminator parameter updates, which can lead to non-convergence [17]—a worse problem than mode collapse. Unrolled GANs [20] propose an optimization that accounts for $k$ gradient steps when computing gradients. We observe that packing achieves better empirical performance with less overhead.

## 2   PacGAN Framework

There are many ways to implement the idea of packing, each with tradeoffs. In this section, we present a simple packing framework that serves as the basis for our empirical experiments and a concrete example of packing. A primary reason for this architectural choice is to emphasize only the effect of packing in numerical experiments, and isolate it from any other effects that might result from other (more sophisticated) changes to the architecture. However, our analysis in Section 3 is agnostic to the packing implementation, and we discuss potential alternative packing architectures in Section 5, especially those that explicitly impose permutation invariance.

We start with an existing GAN, defined by a generator architecture, a discriminator architecture, and a loss function. We call this triplet the *mother architecture*. The PacGAN framework maintains the same generator architecture, loss function, and hyperparameters as the mother architecture. However, instead of using a discriminator $D(X)$ that maps a single sample (either real or generated) to a (soft) label, we use an *augmented* discriminator $D(X_1, X_2, \ldots, X_m)$ that maps $m$ samples to a single (soft) label. These $m$ samples are drawn independently from the same distribution—either real (jointly labelled $Y = 1$) or generated ($Y = 0$). We refer to the concatenation of samples with the same label as *packing*, the resulting discriminator as a *packed discriminator*, and the number $m$ of concatenated samples as the *degree of packing*. The proposed approach can be applied to any existing GAN architecture and any loss function, as long as it uses a discriminator $D(X)$ that classifies a single input sample. We use the notation "Pac(X)(m)" where (X) is the name of the mother architecture, and $(m)$ is is the packing degree. For example, if we take an original GAN and feed the discriminator three packed samples, we call this "PacGAN3".

We implement packing by keeping all hidden layers of the discriminator identical to the mother architecture, and increasing the number of nodes in the input layer by a factor of $m$. For example, in Figure 1, we start with a fully-connected, feed-forward discriminator. Each sample $X$ is two-dimensional, so the input layer has two nodes. Under PacGAN2, we multiply the size of the input layer by the packing degree $m = 2$, and the connections to the first hidden layer are adjusted so that the first two layers remain fully-connected, as in the mother architecture. The grid-patterned nodes in Figure 1 represent input nodes for the second sample. Similarly, when packing a DCGAN, which uses convolutional neural networks for both the generator and the discriminator, we simply stack the images into a tensor of depth $m$. For instance, the discriminator for PacDCGAN4 on the

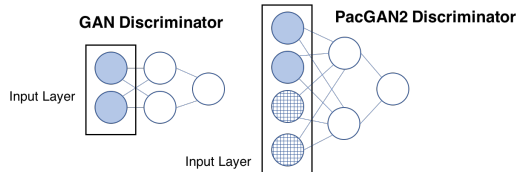

Figure 1: PacGAN(m) augments the input layer by a factor of m. The number of weights between the first two layers are increased to preserve the mother architecture's connectivity. Packed samples are concatenated and fed to the input layer; grid-patterned nodes are input nodes for the second sample.

MNIST dataset of handwritten images [16] would take an input of size $28 \times 28 \times 4$, since each individual black-and-white MNIST image is $28 \times 28$ pixels. Only the input layer and the number of weights in the corresponding first convolutional layer will increase in depth by a factor of 4. As in standard GANs, we train the packed discriminator with a bag of samples from the real data and the generator. However, each minibatch in the stochastic gradient descent now consists of *packed* samples $(X_1, X_2, \ldots, X_m, Y)$, which the discriminator jointly classifies. Intuitively, packing helps the discriminator detect mode collapse because lack of diversity is more obvious in a set of samples than in a single sample.

## 3   Theoretical Analysis of PacGAN

In this section, we show a fundamental connection between the principle of packing and mode collapse in GAN. We provide a complete understanding of how packing changes the loss as seen by the generator, by focusing on $(a)$ the optimal discriminator over a family of all measurable functions; $(b)$ the population expectation; and $(c)$ the 0-1 loss function of the form $\max_D \mathbb{E}_{X \sim P}[\mathbb{I}(D(X))] + \mathbb{E}_{G(Z) \sim Q}[1 - \mathbb{I}(D(G(Z)))]$, subject to $D(X) \in \{0, 1\}$.

This discriminator provides (an approximation of) the total variation distance, and the generator tries to minimize the total variation distance $d_{\mathrm{TV}}(P, Q)$, as widely known in the GAN literature [10]. The reason we make this assumption is primarily for clarity and analytical tractability: total variation distance highlights the effect of packing in a way that is cleaner and easier to understand than if we were to analyze Jensen-Shannon divergence.

We want to understand how this 0-1 loss, as provided by such a discriminator, changes with the *degree of packing* $m$. As packed discriminators see $m$ packed samples, each drawn i.i.d. from one joint class (i.e. either real or generated), we can consider these packed samples as a single sample that is drawn from the product distribution: $P^m$ for real and $Q^m$ for generated. The resulting loss provided by the packed discriminator is therefore $d_{\mathrm{TV}}(P^m, Q^m)$.

We first provide a formal mathematical definition of mode collapse, which leads to a two-dimensional representation of any pair of distributions $(P, Q)$ as a *mode-collapse region*. This region representation provides not only conceptual clarity regarding mode collapse, but also proof techniques that are essential to proving our main results. We defer all the proofs to the Appendix. In Appendix E, we show the proposed mode collapse region is equivalent to the ROC curve for binary hypothesis testing. This allows us to use powerful mathematical techniques from binary hypothesis testing including the data processing inequality.

**Definition 1.** *A target distribution $P$ and a generator $Q$ exhibit $(\varepsilon, \delta)$-mode collapse for $0 \le \varepsilon < \delta \le 1$ if there exists a set $S \subseteq \mathcal{X}$ such that $P(S) \ge \delta$ and $Q(S) \le \varepsilon$.*

Intuitively, larger $\delta$ and smaller $\varepsilon$ indicate more severe mode collapse. That is, if a large portion of the target $P(S) \ge \delta$ in some set $S$ in the domain $\mathcal{X}$ is missing in the generator $Q(S) \le \varepsilon$, we declare $(\varepsilon, \delta)$-mode collapse. Similarly, when we change the role of $P$ and $Q$, and have $P(S) \le \varepsilon$ and $Q(S) \ge \delta$, we say $P$ and $Q$ exhibit $(\epsilon, \delta)$-*mode augmentation*. This definition has a fundamental connection to the ROC region in detection theory and binary hypothesis testing—a connection that is critical for our proof techniques; this connection is detailed in Appendix D and E.

A key observation is that *two pairs of distributions can have the same total variation distance while exhibiting very different mode collapse patterns.* To see this, consider a toy example in Figure 2, with a uniform target distribution $P = U([0, 1])$ and a mode collapsing generator $Q_1 = U([0.2, 1])$

and a non mode collapsing generator $Q_2 = 0.6U([0,0.5]) + 1.4U([0.5,1])$. The appropriate way

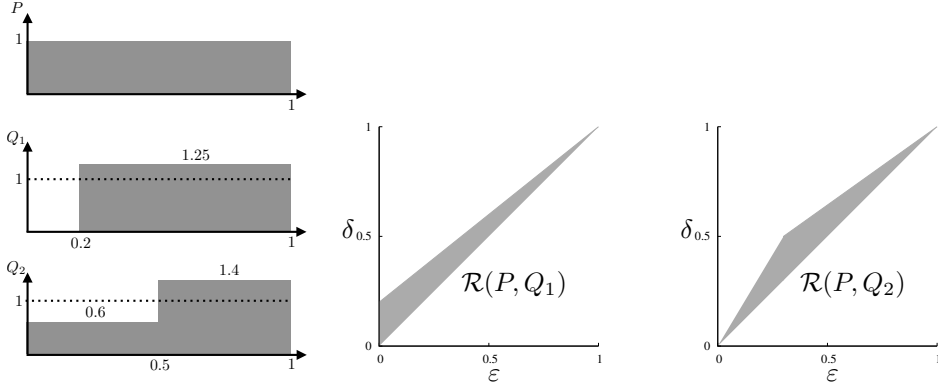

Figure 2: A formal definition of $(\varepsilon, \delta)$-mode collapse and its accompanying region representation captures the intensity of mode collapse for generators $Q_1$ with mode collapse and $Q_2$ which does not have mode collapse, for a toy example distributions $P$, $Q_1$, and $Q_2$ shown on the left. The region of $(\varepsilon, \delta)$-mode collapse that is achievable is shown in grey.

to precisely represent mode collapse is to visualize it through a two-dimensional region we call the *mode collapse region*. For a given pair $(P, Q)$, the corresponding mode collapse region $\mathcal{R}(P, Q)$ is defined as the convex hull of the region of points $(\varepsilon, \delta)$ such that $(P, Q)$ exhibit $(\varepsilon, \delta)$-mode collapse, as shown in Figure 2: $\mathcal{R}(P, Q) \triangleq \mathrm{conv}\big(\{ (\varepsilon, \delta) \mid \delta > \varepsilon \text{ and } (P, Q) \text{ has } (\varepsilon, \delta)\text{-mode collapse} \}\big)$.

There is a fundamental connection between the mode collapse region and the ROC curve in hypothesis testing (Appendix E). An unpacked discriminator, observing only the TV distance between generator distributions $Q$ and the true distribution $P$, cannot distinguish between two candidate generators $Q_1$ and $Q_2$ with $d_{TV}(P, Q_1) = d_{TV}(P, Q_2)$, but different mode collapse regions. The key insight of this work is that by instead considering *product* distributions, the total variation distance $d_{TV}(P^m, Q^m)$ varies in a way that is closely tied to the mode collapse regions for $(P, Q)$. For instance, Figure 3 (left) shows an achievable range of $d_{TV}(P^m, Q^m)$ conditioned on that $d_{TV}(P, Q) = \tau$ for $\tau = 1.1$. Within this achievable range, some pairs $(P, Q)$ have rapidly increasing total variation, occupying the upper part of the region (shown in red, middle panel of Figure 3), and others have slowly increasing total variation, occupying the lower part (shown in blue) in the right panel of Figure 3. We formally show in the following that there is a fundamental connection between total variation distance evolution and degree of mode collapse. Namely, distributions with strong mode collapse occupy the upper region, and hence will be penalized by a packed discriminator.

**Evolution of total variation distances with mode collapse.** We analyze how the total variation evolves for the set of all pairs $(P, Q)$ that have the same total variation distances $\tau$ when unpacked, with $m = 1$, and have $(\varepsilon, \delta)$-mode collapse for some $0 \le \varepsilon < \delta \le 1$. The solution of the following optimization problem gives the desired range:

$$\min_{P,Q} \text{ or } \max_{P,Q} \quad d_{\mathrm{TV}}(P^m, Q^m) \tag{2}$$
$$\text{subject to} \quad d_{\mathrm{TV}}(P, Q) = \tau$$
$$(P, Q) \text{ has } (\varepsilon, \delta)\text{-mode collapse},$$

where the maximization and minimization are over all probability measures $P$ and $Q$, and the mode collapse constraint is defined in Definition 1. We provide the optimal solution analytically and establish that mode-collapsing pairs occupy the upper part of the total variation region; that is, total variation increases rapidly as we pack more samples together (Figure 3, middle panel).

**Theorem 2.** *For all $0 \le \varepsilon < \delta \le 1$ and an integer $m$, if $1 \ge \tau \ge \delta - \varepsilon$ then the solution to the maximization in (2) is $1 - (1 - \tau)^m$, and the solution to the minimization is*

$$\min\Big\{ \min_{0 \le \alpha \le 1 - \frac{\tau\delta}{\delta - \varepsilon}} d_{\mathrm{TV}}\Big( P_{\mathrm{inner1}}(\alpha)^m, Q_{\mathrm{inner1}}(\alpha)^m \Big),$$

$$\min_{1 - \frac{\tau\delta}{\delta - \varepsilon} \le \alpha \le 1 - \tau} d_{\mathrm{TV}}\Big( P_{\mathrm{inner2}}(\alpha)^m, Q_{\mathrm{inner2}}(\alpha)^m \Big) \Big\},$$

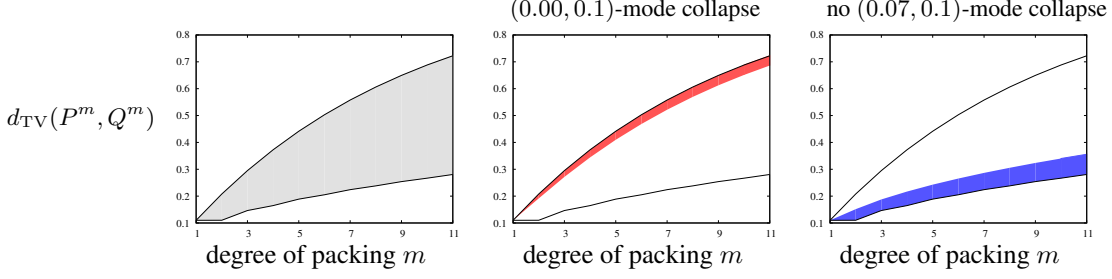

Figure 3: The range of $d_{\mathrm{TV}}(P^m, Q^m)$ achievable by pairs with $d_{\mathrm{TV}}(P,Q) = \tau$, for a choice of $\tau = 0.11$, defined by the solutions of the optimization (4) provided in Theorem 4 in the Appendix (left panel). The range of $d_{\mathrm{TV}}(P^m, Q^m)$ achievable by those pairs that also have $(\varepsilon = 0.00, \delta = 0.1)$-mode collapse (middle panel). A similar range achievable by pairs of distributions that do not have $(\varepsilon = 0.07, \delta = 0.1)$-mode collapse or $(\varepsilon = 0.07, \delta = 0.1)$-mode augmentation (right panel). Pairs $(P, Q)$ with strong mode collapse occupy the top region (near the upper bound) and the pairs with weak mode collapse occupy the bottom region (near the lower bound).

where $P_{\mathrm{inner1}}(\alpha)^m$, $Q_{\mathrm{inner1}}(\alpha)^m$, $P_{\mathrm{inner2}}(\alpha)^m$, and $Q_{\mathrm{inner2}}(\alpha)^m$ are the $m$-th order product distributions of discrete random variables distributed as $P_{\mathrm{inner1}}(\alpha) = [\delta, \quad 1 - \alpha - \delta, \quad \alpha]$, $Q_{\mathrm{inner1}}(\alpha) = [\varepsilon, 1 - \alpha - \tau - \varepsilon, \alpha + \tau]$, $P_{\mathrm{inner2}}(\alpha) = [1 - \alpha, \quad \alpha]$, and $Q_{\mathrm{inner2}}(\alpha) = [1 - \alpha - \tau, \quad \alpha + \tau]$. If $\tau < \delta - \varepsilon$, then the optimization in (2) has no solution and the feasible set is an empty set.

One implication is that distribution pairs $(P, Q)$ at the top of the total variation evolution region are those with the strongest mode collapse. Another implication is that a pair $(P, Q)$ with strong mode collapse (i.e., with larger $\delta$ and smaller $\varepsilon$ in the constraint) will be penalized more under packing, and hence a generator minimizing an approximation of $d_{\mathrm{TV}}(P^m, Q^m)$ will be unlikely to select a distribution that exhibits such strong mode collapse.

**Evolution of total variation distances without mode collapse.** We next analyze how the total variation evolves for the set of all pairs $(P, Q)$ that have the same total variations distances $\tau$ when unpacked, with $m = 1$, and *do not* have $(\varepsilon, \delta)$-mode collapse for some $0 \le \varepsilon < \delta \le 1$. Because of the symmetry of the total variation distance, mode collapse for $(Q, P)$ is equally damaging as mode collapse of $(P, Q)$, when it comes to how fast total variation distances evolve. Hence, we characterize this evolution for those family of pairs of distributions that do not have either mode collapses. The solution of the following optimization problem gives the desired range of total variation distances:

$$\min_{P,Q} \text{ or } \max_{P,Q} \quad d_{\mathrm{TV}}(P^m, Q^m) \tag{3}$$

$$\text{subject to} \quad d_{\mathrm{TV}}(P, Q) = \tau \quad,$$
$$(P, Q) \text{ does not have } (\varepsilon, \delta)\text{-mode collapse,}$$
$$(Q, P) \text{ does not have } (\varepsilon, \delta)\text{-mode collapse,}$$

We provide thte optimal solution analytically and establish that the pairs $(P, Q)$ with weak mode collapse will occupy the bottom part of the evolution of the total variation distances (see Figure 3 right panel), and also will be penalized less under packing. Hence a generator minimizing (approximate) $d_{\mathrm{TV}}(P^m, Q^m)$ is likely to generate distributions with weak mode collapse.

**Theorem 3.** *If $\delta + \varepsilon \le 1$ and $\delta - \varepsilon \le \tau \le (\delta - \varepsilon)/(\delta + \varepsilon)$ then the solution to the maximization in (3) is*

$$\max_{\alpha + \beta \le 1 - \tau, \frac{\varepsilon \tau}{\delta - \varepsilon} \le \alpha, \beta} \quad d_{\mathrm{TV}}\Big( P_{\mathrm{outer1}}(\alpha, \beta)^m, Q_{\mathrm{outer1}}(\alpha, \beta)^m \Big),$$

*where $P_{\mathrm{outer1}}(\alpha, \beta)^m$ and $Q_{\mathrm{outer1}}(\alpha, \beta)^m$ are the $m$-th order product distributions of discrete random variables distributed as $P_{\mathrm{outer1}}(\alpha, \beta) = [\frac{\alpha(\delta - \varepsilon) - \varepsilon\tau}{\alpha - \varepsilon}, \frac{\alpha(\alpha + \tau - \delta)}{\alpha - \varepsilon}, 1 - \tau - \alpha - \beta, \beta, 0]$ and $Q_{\mathrm{outer1}}(\alpha, \beta) = [0, \alpha, 1 - \tau - \alpha - \beta, \frac{\beta(\beta + \tau - \delta)}{\beta - \varepsilon}, \frac{\beta(\delta - \varepsilon) - \varepsilon\tau}{\beta - \varepsilon}]$. The solution to the minimization in (3) is*

$$\min_{\frac{\varepsilon \tau}{\delta - \varepsilon} \le \alpha \le 1 - \frac{\delta\tau}{\delta - \varepsilon}} \quad d_{\mathrm{TV}}\Big( P_{\mathrm{inner}}(\alpha)^m, Q_{\mathrm{inner}}(\alpha, \tau)^m \Big),$$

*where $P_{\text{inner}}(\alpha)$ and $Q_{\text{inner}}(\alpha, \tau)$ are defined as in Theorem 4 in the Appendix.*

We can prove the exact solution of the optimization for all values of $\varepsilon$ and $\delta$, which we provide in the Appendix. We refer also to the appendix of more illustrations of regions occupied by various choices of $\varepsilon$ and $\delta$ for mode collapsing distributions, and non mode collapsing regions.

## 4    Experiments

On standard benchmark datasets, we compare PacGAN to several baseline GAN architectures, some explicitly designed to mitigate mode collapse: GAN [10], minibatch discrimination (MD) [24], DCGAN [22], VEEGAN [25], Unrolled GANs [20], and ALI [8]. We also implicitly compare against BIGAN [7], which is conceptually identical to ALI. To isolate the effects of packing, we make minimal choices in the architecture and hyperparameters of our packing implementation. Our goal is to reproduce experiments from the literature, apply packing to the simplest baseline GAN, and observe how packing affects performance. Whenever possible, we use the exactly same choice of architecture, hyperparameters, and loss function as a baseline in each experiment; we change only the discriminator to accept packed samples. All code to reproduce our experiments can be found at `https://github.com/fjxmlzn/PacGAN`.

**Metrics.** We measure several previously-used metrics. The first is *number of modes* that are produced by a generator [7, 20, 25]. In labelled datasets, this number can be evaluated using a third-party trained classifier that classifies the generated samples [25]. A second metric used in [25] is the *number of high-quality samples*, which is the proportion of the samples that are within $x$ standard deviations from the center of a mode. Finally, we measure the *reverse Kullback-Leibler divergence* between the induced distribution from generated samples and the induced distribution from the real samples. Each of these metrics has shortcomings—for example, the number of observed modes ignores class imbalance, and all of the metrics assume the modes are known a priori.

**Datasets.** We use synthetic and real datasets. The **2D-ring** [25] is a mixture of eight two-dimensional spherical Gaussians with means $(\cos((2\pi/8)i), \sin((2\pi/8)i))$ and variances $10^{-4}$ in each dimension for $i \in \{1, \ldots, 8\}$. The **2D-grid** [25] is a mixture of 25 two-dimensional spherical Gaussians with means $(-4 + 2i, -4 + 2j)$ and variances $0.0025$ in each dimension for $i, j \in \{0, 1, 2, 3, 4\}$. The MNIST dataset [16] consists of 70K images of handwritten digits, each $28 \times 28$ pixels. Unmodified, this dataset has 10 modes (digits). As in [20, 25], we augment the number of modes by *stacking* the images: we generate a new dataset of 128K images where each image consists of three random MNIST images stacked into a $28 \times 28 \times 3$ RGB image. This new **stacked MNIST** dataset has (with high probability) $1000 = 10 \times 10 \times 10$ modes. Finally, we include experiments on the **CelebA** dataset, which is a collection of 200K facial images of celebrities [19].

### 4.1    Synthetic data experiments

Our first experiment measures the effect of the number of discriminator parameters on mode collapse. Packed architectures have more discriminator nodes (and parameters) than the mother architecture, which could artificially mitigate mode collapse by giving the discriminator more capacity. We compare this effect to the effect of packing on the 2D grid dataset. In Figure 4, we evaluate the number of modes recovered and reverse KL divergence for ALI, GAN, MD, and PacGAN, while varying the number of *total* parameters in each architecture (discriminator and encoder if one exists). The experimental details are included in Appendix A.2. For MD, the metrics first improve and then degrade with the number of parameters. We suspect that this may because MD is very sensitive to experiment settings, as the same architecture of MD has very different performance on 2d-grid and 2d-ring dataset (Appendix A.1). For ALI, GAN and PacGAN, despite varying the number of parameters by an order of magnitude, we do not see significant evidence of the metrics improving with the number of parameters. This suggests that the advantages of PacGAN and ALI compared to GAN do not stem from having more parameters. Packing also seems to increase the number of modes recovered and decrease the reverse KL divergence; we explore this phenomenon more in subsequent experiments.

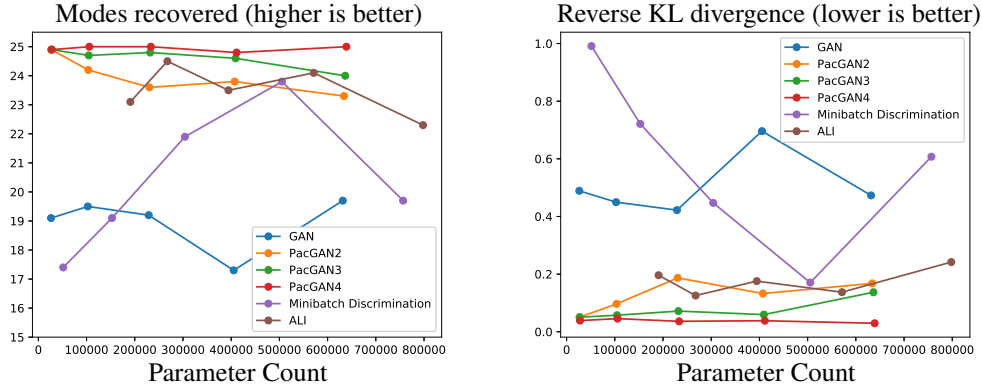

Figure 4: Effect of number of parameters on evaluation metrics.

## 4.2 Stacked MNIST experiments

For our stacked MNIST experiments, we generate samples. Each of the three channels in each sample is classified by a pre-trained third-party MNIST classifier, and the resulting three digits determine which of the 1000 modes the sample belongs to. We measure the number of modes captured, as well as the KL divergence between the generated distribution over modes and the expected (uniform) one.

In the first experiment, we replicate Table 2 from [25], which measured the number of observed modes in a generator trained on the stacked MNIST dataset, as well as the KL divergence of the generated mode distribution. In line with [25], we used a DCGAN-like architecture for these experiments[1] (details in Appendix B.1). Our results are shown in Table 1. The first four rows are copied directly from [25]. The last three rows are computed using a basic DCGAN, with packing in the discriminator. We find that packing gives good mode coverage, reaching all 1,000 modes in every trial. Again, packing the simplest DCGAN fully captures all the modes in the benchmark test, so we do not pursue packing more complex baseline architectures. We also observe that MD is very unstable throughout training, which makes it capture even less modes than GAN. One factor that contributes to MD's instability may be that MD requires too many parameters. The number of discriminator parameters in MD is 47,976,773, whereas GAN has 4,310,401 and PacGAN4 only needs 4,324,801.

|  | Stacked MNIST | |
|---|---|---|
|  | Modes | KL |
| DCGAN [22] | 99.0 | 3.40 |
| ALI [8] | 16.0 | 5.40 |
| Unrolled GAN [20] | 48.7 | 4.32 |
| VEEGAN [25] | 150.0 | 2.95 |
| Minibatch Discrimination [24] | $24.5\pm7.67$ | $5.49\pm0.418$ |
| DCGAN (our implementation) | $78.9\pm6.46$ | $4.50\pm0.127$ |
| PacDCGAN2 (ours) | $1000.0\pm0.00$ | $0.06\pm0.003$ |
| PacDCGAN3 (ours) | $1000.0\pm0.00$ | $0.06\pm0.003$ |
| PacDCGAN4 (ours) | $1000.0\pm0.00$ | $0.07\pm0.005$ |

Table 1: Two measures of mode collapse proposed in [25] for the stacked MNIST dataset: number of modes captured by the generator and reverse KL divergence over the generated mode distribution. The DCGAN, PacDCGAN, and MD results are averaged over 10 trials, with standard error reported.

## 4.3 CelebA experiments

Finally, we measure the diversity of images generated from the CelebA dataset as in [3] by estimating the probability of collision in a batch of generated images. If there exists at least one pair of near-duplicate images in the batch, a collision is declared, which indicates lack of diversity. The details of

how we determine duplicates and our architecture are deferred to Appendix C. We find that packing significantly improves the diversity of samples, and if the size of the discriminator is small, packing also improves sample quality. See Appendix C for generated samples.

| discriminator size | probability of collision | |
| --- | --- | --- |
| | DCGAN | PacDCGAN2 |
| 273K | 1 | 0.33 |
| $4 \times 273\text{K}$ | 0.42 | 0 |
| $16 \times 273\text{K}$ | 0.86 | 0 |
| $25 \times 273\text{K}$ | 0.65 | 0.17 |

Table 2: Probability of $\geq 1$ pair of near-duplicate images appearing in a batch of 1024 images generated from DCGAN and PacDCGAN2 on celebA dataset.

## 5    Discussion

In this work, we propose a packing framework that theoretically and empirically mitigates mode collapse with low overhead. Our analysis leads to several interesting open questions, including how to apply these analysis techniques to more general classes of loss functions such as Jensen-Shannon divergence and Wasserstein distances. This will complete the understanding of the superiority of our approach observed in experiments with JS divergence in Section 4 and with Wasserstein distance in Appendix B.3. Another important question is what packing architecture to use. For instance, a framework that provides permutation invariance may give better results such as graph neural networks [6, 26, 15] or deep sets [28].

## Acknowledgement

The authors would like to thank Sreeram Kannan and Alex Dimakis for the initial discussions that lead to the inception of the packing idea, and Vyas Sekar for valuable discussions about GANs. We thank Srivastava Akash, Luke Metz, Tu Nguyen, and Yingyu Liang for providing insights and/or the implementation details on their proposed architectures for VEEGAN [25], Unrolled GAN [20], D2GAN [21], and MIX+GAN [2]. We thank the anonymous reviewers for their constructive feedback.

This work is supported by NSF awards CNS-1527754, CCF-1553452, CCF-1705007, and RI-1815535 and Google Faculty Research Award. This work used the Extreme Science and Engineering Discovery Environment (XSEDE), which is supported by National Science Foundation grant number OCI-1053575. Specifically, it used the Bridges system, which is supported by NSF award number ACI-1445606, at the Pittsburgh Supercomputing Center (PSC). This work is partially supported by the generous research credits on AWS cloud computing resources from Amazon.

## Footnotes

[1] https://github.com/carpedm20/DCGAN-tensorflow

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
