[Supplementary Material]

# Appendix

## A  Synthetic data experiments

We ran two experiments on synthetic data. In the first, we evaluate mode collapse for PacGAN, GANs, ALI, and MD in order to reproduce Table 1 from [25]. In the second, we evaluate mode collapse as a function of the number of GAN parameters. The second experiment is described in Section 4.1 of the main paper.

### A.1  Experiment 1: Reproduce Table 1 from [25]

Our first experiment evaluates the number of modes and the number of high-quality samples for the 2D-ring and the 2D-grid. The hyperparameters, network architecture, and loss function for GAN and ALI are exactly reproduced from ALI's architecture[2]. PacGAN is adapted from GAN by changing the dimension of input layer in discriminator without any further hyper-parameter tuning.

All of the GANs in this experiment use the same generator architecture. There are four hidden layers, each of which has 400 units with ReLU activation, trained with batch normalization [11]. The input noise is a two dimensional spherical Gaussian with zero mean and unit variance. GAN, ALI, and PacGAN in this experiment use the same discriminator, except that the input dimensions are different. The discriminator has three hidden layers, with 200 units per hidden layer. The hidden layers use LinearMaxout with 5 maxout pieces, and no batch normalization is used in the discriminator. In addition to a generator and discriminator, ALI also has a third component, called an *encoder*; we only used the encoder to evaluate ALI, but did not include the encoder in our PacGAN. MD's discriminator is the same as GAN's discriminator, except that a minibatch discrimination layer is added before the output layer. The implementation of minibatch discrimination layer in this and all following experiments is based on the standard implementation[3].

We train each GAN with 100,000 total samples, and a mini-batch size of 100 samples; training is run for 400 epochs. The discriminator's loss function is $\log(1 + \exp(-D(\text{real data}))) + \log(1 + \exp(D(\text{generated data})))$. The generator's loss function is $\log(1 + \exp(D(\text{real data}))) + \log(1 + \exp(-D(\text{generated data})))$. Adam [14] stochastic gradient descent is applied with the generator weights and the discriminator weights updated once per mini-batch. At testing, we use 2500 samples from the learned generator for evaluation.

Figure 5 shows the plots from GAN and PacGAN2 in 2D grid experiment.

| | 2D-ring | | | 2D-grid | | |
|---|---|---|---|---|---|---|
| | Modes (Max 8) | high quality samples | reverse KL | Modes (Max 25) | high quality samples | reverse KL |
| GAN [10] | 6.3±0.5 | 98.2±0.2 % | 0.45±0.09 | 17.3±0.8 | 94.8±0.7 % | 0.70±0.07 |
| ALI [8] | 6.6±0.3 | 97.6±0.4 % | 0.36±0.04 | 24.1±0.4 | 95.7±0.6 % | 0.14±0.03 |
| Minibatch Disc. [24] | 4.3±0.8 | 36.6±8.8 % | 1.93±0.11 | 23.8±0.5 | 79.9±3.2 % | 0.17±0.03 |
| PacGAN2 (ours) | 7.9±0.1 | 95.6±2.0 % | 0.07±0.03 | 23.8±0.7 | 91.3±0.8 % | 0.13±0.04 |
| PacGAN3 (ours) | 7.8±0.1 | 97.7±0.3 % | 0.10±0.02 | 24.6±0.4 | 94.2±0.4 % | 0.06±0.02 |
| PacGAN4 (ours) | 7.8±0.1 | 95.9±1.4 % | 0.07±0.02 | 24.8±0.2 | 93.6±0.6 % | 0.04±0.01 |

Table 3: Two measures of mode collapse proposed in [25] for two synthetic mixtures of Gaussians: number of modes captured by the generator and percentage of high quality samples, as well as reverse KL. Note that 2 trials of MD in 2D-ring dataset cover no mode, which makes reverse KL intractable. This reverse KL entry is averaged over the other 8 trails. All other results are averaged over 10 trials, with standard error reported.

Our results (Table 3) show that PacGAN outperforms or matches the baseline schemes. On the 2D-grid, increasing the packing degree $m$ increases the average number of modes recovered, as expected. On the 2D-ring, PacGAN2 recovers almost all the modes, so further packing provides

little extra benefit. Regardless, packing significantly reduces mode collapse compared to the mother architecture.

We also observe that in terms of mode coverage MD performs well in 2D-grid dataset but badly in 2D-ring dataset, even with completely the same architecture. This suggests that MD is sensitive to experiment settings. In terms of high quality samples, MD performs even worse than GAN baseline in both datasets.

Figure 5: Scatter plot of the 2D samples from the true distribution (left) of 2D-grid and the learned generators using GAN (middle) and PacGAN2 (right). PacGAN2 captures all of the 25 modes.

### A.2   Mode collapse vs. number of parameters (Section 4.1)

In this experiment, we keep the input and output layers identical to our experiment in Appendix A.1, but alter the number of nodes per hidden layer in the discriminator. For each experimental setting, each hidden layer of the discriminator has the same number of hidden nodes, drawn from the set $\{50, 100, 150, 200, 250\}$. This hidden layer size determines the total number of parameters in the architecture, so each GAN variant is evaluated for five different parameters. Each data point is averaged over 10 trials. The horizontal axis shows the total number of parameters in the discriminator and encoder (if one exists).

## B   Stacked MNIST Experiments

We ran two experiments on the stacked MNIST dataset. The first is reproducing an experiment from VEEGAN paper, as described in Section 4.2. The second is reproducing an experiment from the Unrolled GANs paper.

### B.1   Generated pictures and architecture details from VEEGAN experiment

The generator architecture and discriminator architecture of GAN and PacGAN are shown in Table 4 and Table 5 respectively. MD uses the same architecture as GAN, except that a minibatch discrimination layer is added before the output layer of discriminator. We train each GAN on 128,000 samples, with a mini-batch size of 64. The generator's loss function is $-\log(D(\text{generated data}))$, and the discriminator's loss function is -log(D(real data))-log(1-D(generated data)). We update the generator parameters twice and the discriminator parameters once in each mini-batch, and train the networks over 50 epochs. For testing, we generate 26,000 samples, and evaluate the empirical KL divergence and number of modes covered. Finally, we average these values over 10 runs of the entire pipeline.

The generated pictures of DCGAN and PacDCGAN2 are shown in Figure 6.

### B.2   Unrolled GAN experiment

Unrolled GANs exploit the observation that iteratively updating discriminator and generator model parameters can contribute to training instability. To mitigate this, they update model parameters by computing the loss function's gradient with respect to $k \geq 1$ sequential discriminator updates, where

| layer | number of outputs | kernel size | stride | BN | activation function |
|---|---|---|---|---|---|
| Input: $z \sim U(-1,1)^{100}$ | 100 | | | | |
| Fully connected | 2*2*512 | | | Yes | ReLU |
| Transposed Convolution | 4*4*256 | 5*5 | 2 | Yes | ReLU |
| Transposed Convolution | 7*7*128 | 5*5 | 2 | Yes | ReLU |
| Transposed Convolution | 14*14*64 | 5*5 | 2 | Yes | ReLU |
| Transposed Convolution | 28*28*3 | 5*5 | 2 | | Tanh |

Table 4: Generator architecture in VEEGAN experiment

| layer | number of outputs | kernel size | stride | BN | activation function |
|---|---|---|---|---|---|
| Input: $x \sim p_{data}^m$ or $G^m$ | 28*28*(3*m) | | | | |
| Convolution | 14*14*64 | 5*5 | 2 | | LeakyReLU |
| Convolution | 7*7*128 | 5*5 | 2 | Yes | LeakyReLU |
| Convolution | 4*4*256 | 5*5 | 2 | Yes | LeakyReLU |
| Convolution | 2*2*512 | 5*5 | 2 | Yes | LeakyReLU |
| Fully connected | 1 | | | | Sigmoid |

Table 5: Discriminator architecture in VEEGAN experiment ($m$ stands for the degree of packing)

Target distribution          DCGAN          PacDCGAN2

Figure 6: True distribution (left), DCGAN generated samples (middle), and PacDCGAN2 generated samples (right) from the stacked-MNIST dataset show PacDCGAN2 captures more diversity while producing sharper images.

$k$ is called the unrolling parameter. [20] reports that unrolling improves mode collapse as $k$ increases, at the expense of greater training complexity.

Unlike Section 4.2, this experiment studies the effect of the discriminator size on the number of modes learned by a generator. The key differences between these trials and the unrolled GAN row in Table 1 are four: (1) the unrolling parameters are different, (2) the discriminator sizes are different, (3) the generator and discriminator architectures are chosen according to Appendix E in [20], and (4) the total training time was 5 times longer than the one used in [20]. PacDCGAN uses the same generators and discriminators (except for input layer) as unrolled GAN in each experiment. MD uses the same architecture, except that a minibatch discrimination layer is added before the output layer of discriminator.

Our results are reported in Table 6. The first four rows are copied from [20]. As before, we find that packing seems to increase the number of modes covered. Additionally, in both experiments, PacDCGAN finds more modes on average than Unrolled GANs with $k = 10$, with lower reverse KL divergences between the mode distributions. This suggests that packing has a more pronounced effect than unrolling.

We see that compared with PacGAN, MD has worse metrics in D=1/4G setting but has similar metrics in D=1/2G setting. In addition, we should note that MD requires much more discriminator parameters: 747 for PacGAN4 and 1,226,317 for MD in D=1/4G setting; 2,213 for PacGAN4 and 2,458,533 for MD in D=1/2G setting.

|  | D is $1/4$ size of G | | D is $1/2$ size of G | |
| --- | --- | --- | --- | --- |
|  | Modes (Max 1000) | KL | Modes (Max 1000) | KL |
| DCGAN [22] | 30.6±20.73 | 5.99±0.42 | 628.0±140.9 | 2.58±0.75 |
| Unrolled GAN, 1 step [20] | 65.4±34.75 | 5.91±0.14 | 523.6±55.77 | 2.44±0.26 |
| Unrolled GAN, 5 steps [20] | 236.4±63.30 | 4.67±0.43 | 732.0±44.98 | 1.66±0.09 |
| Unrolled GAN, 10 steps [20] | 327.2±74.67 | 4.66±0.46 | 817.4±37.91 | 1.43±0.12 |
| Minibatch Discrimination [24] | 264.1±59.02 | 3.32±0.30 | 837.1±67.46 | 0.84±0.25 |
| DCGAN (our implementation) | 78.5±17.56 | 5.21±0.19 | 487.7±34.59 | 2.24±0.15 |
| PacDCGAN2 (ours) | 484.5±32.99 | 2.61±0.22 | 840.7±15.92 | 1.00±0.05 |
| PacDCGAN3 (ours) | 601.3±32.18 | 2.00±0.17 | 866.6±12.10 | 0.90±0.04 |
| PacDCGAN4 (ours) | 667.4±29.00 | 1.81±0.15 | 820.2±25.50 | 1.15±0.14 |

Table 6: Modes covered and KL divergence for unrolled GANs and MD as compared to PacDCGAN for various unrolling parameters, discriminator sizes, and the degree of packing. The DCGAN and PacDCGAN results are averaged over 50 trials, with standard error reported. The MD results are averaged over 10 trials, with standard error reported.

### B.3 Comparisons to WGAN

To verify that our packing idea can also work on Wasserstein loss, we compare WGAN with PacWGAN on stacked MNIST dataset. The experiment setting follows Appendix B.1, except: (1) remove all batch normalization layers in discriminator, and remove the sigmoid activation in the output layer of discriminator, (2) use WGAN-GP loss instead of JSD loss, (3) to showcase the difference between WGAN and PacGAN, we use smaller generators and discriminators. Specifically, the number of feature maps in each layer of discriminator and generator is a quarter of what used in Appendix B.1.

Table 7 shows the results. We find that PacWGANs discover more modes and achieves smaller KL than WGAN. This suggests that packing can also work on Wasserstein loss.

|  | Stacked MNIST | |
| --- | --- | --- |
|  | Modes | KL |
| WGAN [1] | 314.3±38.54 | 2.44±0.170 |
| PacWGAN2 (ours) | 927.6±22.19 | 0.59±0.108 |
| PacWGAN3 (ours) | 948.7±21.43 | 0.50±0.089 |
| PacWGAN4 (ours) | 965.7±19.07 | 0.42±0.094 |

Table 7: Two measures of mode collapse proposed in [25] for the stacked MNIST dataset: number of modes captured by the generator and reverse KL divergence over the generated mode distribution. All results are averaged over 10 trials, with standard error reported.

## C  Generated pictures in CelebA experiment

To detect collisions in a batch of images, [3] selects the 20 closest generated sample pairs (measured in Euclidean distance over pixel space), and then manually determine if any of the pairs would be considered duplicates by humans. We estimate the probability of collision by repeating the experiment 10 times and averaging over three human reviewers on each sample batch. We use DCGAN- unconditional, with JSD objective as described in [22] as the base architecture. We perform the experiment for different sizes of the discriminator while fixing the other hyper-parameters. It has 4 CNN layers in the discriminator with the number of output channels of each layer being $dim \times 1, 2, 4, 8$. Thus the discriminator size is proportional to $dim^2$. Table 2 shows probability of collision in a batch of size 1024 for DCGAN and PacDCGAN2 for $dim \in \{16, 32, 64, 80\}$.

Figure 7 shows samples generated from DCGAN and PacDCGAN2 for $dim = 16$. We note that DCGAN and PacDCGAN2 use approximately same number of parameters, 273K and 274K respectively.

DCGAN                    PacDCGAN2

Figure 7: CelebA samples generated from DCGAN (left) and PacDCGAN2 (right) show PacDCGAN2 generates more diverse and sharper images.

# D  Theoretical understanding of mode collapse and product distributions

## D.1  Evolution of the region under product distributions

The toy example generators $Q_1$ and $Q_2$ from Figure 2 could not be distinguished using only their total variation distances from $P$, despite exhibiting very different mode collapse properties. This suggests that the original GAN (with 0-1 loss) may be vulnerable to mode collapse. We prove in Theorem 2 that a discriminator that packs multiple samples together *can* better distinguish mode-collapsing generators. Intuitively, $m$ packed samples are equivalent to a single sample drawn from the product distributions $P^m$ and $Q^m$. We show in this section that there is a fundamental connection between the strength of mode collapse of $(P, Q)$ and the loss as seen by the packed discriminator $d_{TV}(P^m, Q^m)$.

**Intuition via toy examples.** Concretely, consider the example from the previous section and recall that $P^m$ denote the product distribution resulting from packing together $m$ independent samples from $P$. Figure 8 illustrates how the mode collapse region evolves over $m$, the degree of packing. This evolution highlights a key insight: the region $\mathcal{R}(P^m, Q_1^m)$ of a mode-collapsing generator expands much faster as $m$ increases compared to the region $\mathcal{R}(P^m, Q_2^m)$ of a non-mode-collapsing generator. This implies that the total variation distance of $(P, Q_1)$ increases more rapidly as we pack more samples, compared to $(P, Q_2)$. This follows from the fact that the total variation distance between $P$ and the generator can be determined directly from the upper boundary of the mode collapse region (see Section E.2 for the precise relation). In particular, a larger mode collapse region implies a larger total variation distance between $P$ and the generator, which is made precise in Section E.2. The total variation distances $d_{TV}(P^m, Q_1^m)$ and $d_{TV}(P^m, Q_2^m)$, which were explicitly chosen to be equal at $m = 1$ in our example, grow farther apart with increasing $m$, as illustrated in the right figure below. This implies that if we use a packed discriminator, the mode-collapsing generator $Q_1$ will be heavily penalized for having a larger loss, compared to the non-mode-collapsing $Q_2$.

Figure 8: Evolution of the mode collapse region over the degree of packing $m$ for the two toy examples from Figure 2. The region of the mode-collapsing generator $Q_1$ expands faster than the non-mode-collapsing generator $Q_2$ when discriminator inputs are packed (at $m = 1$ these examples have the same TV distances). This causes a discriminator to penalize mode collapse as desired.

## D.2 Evolution of total variation distances

In order to generalize the intuition from the above toy examples, we first analyze how the total variation evolves for the set of all pairs $(P, Q)$ that have the same total variation distance $\tau$ when unpacked (i.e., when $m = 1$). The solutions to the following optimization problems give the desired upper and lower bounds, respectively, on total variation distance for any distribution pair in this set with a packing degree of $m$:

$$\min_{P,Q} \; d_{\text{TV}}(P^m, Q^m) \qquad\qquad \max_{P,Q} \; d_{\text{TV}}(P^m, Q^m) \qquad (4)$$
$$\text{subject to} \;\; d_{\text{TV}}(P, Q) = \tau \qquad\qquad \text{subject to} \;\; d_{\text{TV}}(P, Q) = \tau \;,$$

where the maximization and minimization are over all probability measures $P$ and $Q$. We give the exact solution in Theorem 4, which is illustrated pictorially in Figure 3 (left).

**Theorem 4.** *For all $0 \leq \tau \leq 1$ and a positive integer $m$, the solution to the maximization in* (4) *is $1 - (1 - \tau)^m$, and the solution to the minimization in* (4) *is*

$$L(\tau, m) \quad \triangleq \quad \min_{0 \leq \alpha \leq 1 - \tau} \; d_{\text{TV}}\Big( P_{\text{inner}}(\alpha)^m, Q_{\text{inner}}(\alpha, \tau)^m \Big) \;, \qquad (5)$$

*where $P_{\text{inner}}(\alpha)^m$ and $Q_{\text{inner}}(\alpha, \tau)^m$ are the $m$-th order product distributions of binary random variables distributed as*

$$P_{\text{inner}}(\alpha) \quad = \quad [1 - \alpha, \quad \alpha] \;, \qquad (6)$$
$$Q_{\text{inner}}(\alpha, \tau) \quad = \quad [1 - \alpha - \tau, \quad \alpha + \tau] \;. \qquad (7)$$

Although this is a simple statement that can be proved in several different ways, we introduce in Section F a novel geometric proof technique that critically relies on the proposed mode collapse region. This particular technique will allow us to generalize the proof to more complex problems involving mode collapse in Theorem 2, for which other techniques do not generalize. Note that the claim in Theorem 4 has nothing to do with mode collapse. Still, the mode collapse region definition (used here purely as a proof technique) provides a novel technique that seamlessly generalizes to prove more complex statements in the following.

For any given value of $\tau$ and $m$, the bounds in Theorem 4 are easy to evaluate numerically, as shown below in the left panel of Figure 3. Within this achievable range, some subset of pairs $(P, Q)$ have rapidly increasing total variation, occupying the upper part of the region (shown in red, middle panel of Figure 3), and some subset of pairs $(P, Q)$ have slowly increasing total variation, occupying the lower part as shown in blue in the right panel in Figure 3. In particular, the evolution of the mode-collapse region of a pair of $m$-th power distributions $\mathcal{R}(P^m, Q^m)$ is fundamentally connected to the strength of mode collapse in the original pair $(P, Q)$. This means that for a mode-collapsed pair $(P, Q_1)$, the $m$th-power distribution will exhibit a different total variation distance evolution than a non-mode-collapsed pair $(P, Q_2)$. As such, these two pairs can be distinguished by a packed discriminator. Making such a claim precise for a broad class of mode-collapsing and non-mode-collapsing generators is challenging, as it depends on the target $P$ and the generator $Q$, each of which can be a complex high dimensional distribution, like natural images. The proposed region interpretation, endowed with the hypothesis testing interpretation and the data processing inequalities that come with it, is critical: it enables the abstraction of technical details and provides a simple and tight proof based on *geometric techniques* on two-dimensional regions.

## D.3 Evolution of total variation distances with mode collapse

We plot the region defined by the upper and lower bounds in Theorem 2 for various values of $\varepsilon$ and $\delta$.

We repeat the main theorem here for completeness.

**Theorem 5.** *For all $0 \leq \varepsilon < \delta \leq 1$ and a positive integer $m$, if $1 \geq \tau \geq \delta - \varepsilon$ then the solution to the maximization in* (2) *is $1 - (1 - \tau)^m$, and the solution to the minimization in* (2) *is*

$$L_1(\varepsilon, \delta, \tau, m) \quad \triangleq \quad \min \Big\{ \min_{0 \leq \alpha \leq 1 - \frac{\tau\delta}{\delta - \varepsilon}} \; d_{\text{TV}}\Big( P_{\text{inner1}}(\delta, \alpha)^m, Q_{\text{inner1}}(\varepsilon, \alpha, \tau)^m \Big) ,$$

$$\min_{1 - \frac{\tau\delta}{\delta - \varepsilon} \leq \alpha \leq 1 - \tau} \; d_{\text{TV}}\Big( P_{\text{inner2}}(\alpha)^m, Q_{\text{inner2}}(\alpha, \tau)^m \Big) \Big\} \;, \qquad (8)$$

where $P_{\mathrm{inner1}}(\delta, \alpha)^m$, $Q_{\mathrm{inner1}}(\varepsilon, \alpha, \tau)^m$, $P_{\mathrm{inner2}}(\alpha)^m$, and $Q_{\mathrm{inner2}}(\alpha, \tau)^m$ are the m-th order product distributions of discrete random variables distributed as

$$
\begin{aligned}
P_{\mathrm{inner1}}(\delta, \alpha) &= [\delta, \quad 1 - \alpha - \delta, \quad \alpha] \,, & (9) \\
Q_{\mathrm{inner1}}(\varepsilon, \alpha, \tau) &= [\varepsilon, \quad 1 - \alpha - \tau - \varepsilon, \quad \alpha + \tau] \,, & (10) \\
P_{\mathrm{inner2}}(\alpha) &= [1 - \alpha, \quad \alpha] \,, & (11) \\
Q_{\mathrm{inner2}}(\alpha, \tau) &= [1 - \alpha - \tau, \quad \alpha + \tau] \,. & (12)
\end{aligned}
$$

*If $\tau < \delta - \varepsilon$, then the optimization in (2) has no solution and the feasible set is an empty set.*

A proof of this theorem is provided in Section F.2, which critically relies on the proposed mode collapse region representation of the pair $(P, Q)$, and the celebrated result by Blackwell from [4]. The solutions in Theorem 2 can be numerically evaluated for any given choices of $(\varepsilon, \delta, \tau)$ as we show in Figure 9.

Figure 9: The evolution of total variation distance over the packing degree $m$ for mode collapsing pairs is shown as a red band. The upper and lower boundaries of the red band is defined by the optimization 2 and computed using Theorem 2. For a fixed $d_{\mathrm{TV}}(P, Q) = \tau = 0.11$ and $(\varepsilon, \delta = 0.1)$-mode collapse, we show the evolution with different choices of $\varepsilon \in \{0.00, 0.01, 0.02, 0.03, 0.04, 0.05\}$. The black solid lines show the maximum/minimum total variation in the optimization problem (4) as a reference. The family of pairs $(P, Q)$ with stronger mode collapse (i.e. smaller $\varepsilon$ in the constraint), occupy a smaller region at the top with higher total variation under packing, and hence is more penalized when training the generator.

### D.4 Evolution of total variation distances without mode collapse

For the optimization in (3), it is not possible to have $d_{\mathrm{TV}}(P, Q) > (\delta - \varepsilon)/(\delta + \varepsilon)$ and $\delta + \varepsilon \leq 1$, and satisfy the mode collapse and mode augmentation constraints (see Section F.3 for a proof). Similarly, it is not possible to have $d_{\mathrm{TV}}(P, Q) > (\delta - \varepsilon)/(2 - \delta - \varepsilon)$ and $\delta + \varepsilon \geq 1$, and satisfy the constraints. Hence, the feasible set is empty when $\tau > \max\{(\delta - \varepsilon)/(\delta + \varepsilon), (\delta - \varepsilon)/(2 - \delta - \varepsilon)\}$. On the other hand, when $\tau \leq \delta - \varepsilon$, no pairs with total variation distance $\tau$ can have $(\varepsilon, \delta)$-mode collapse. In this case, the optimization reduces to the simpler one in (4) with no mode collapse constraints. Non-trivial solution exists in the middle regime, i.e. $\delta - \varepsilon \leq \tau \leq \max\{(\delta - \varepsilon)/(\delta + \varepsilon), (\delta - \varepsilon)/(2 - \delta - \varepsilon)\}$. The lower bound for this regime, given in equation (16), is the same as the lower bound in (5), except it optimizes over a different range of $\alpha$ values. For a wide range of parameters $\varepsilon$, $\delta$, and $\tau$, those lower bounds will be the same, and even if they differ for some parameters, they differ slightly. This implies that the pairs $(P, Q)$ with weak mode collapse will occupy the bottom part of the evolution of the total variation distances (see Figure 3 right panel), and also will be penalized less under packing.

Hence a generator minimizing (approximate) $d_{\mathrm{TV}}(P^m, Q^m)$ is likely to generate distributions with weak mode collapse.

**Theorem 6.** *For all $0 \le \varepsilon < \delta \le 1$ and a positive integer $m$, if $0 \le \tau < \delta - \varepsilon$, then the maximum and the minimum of (3) are the same as those of the optimization (4) provided in Theorem 4.*

*If $\delta + \varepsilon \le 1$ and $\delta - \varepsilon \le \tau \le (\delta - \varepsilon)/(\delta + \varepsilon)$ then the solution to the maximization in (3) is*

$$U_1(\epsilon, \delta, \tau, m) \quad \triangleq \quad \max_{\alpha + \beta \le 1 - \tau, \frac{\varepsilon\tau}{\delta - \varepsilon} \le \alpha, \beta} \; d_{\mathrm{TV}}\Big( P_{\mathrm{outer1}}(\varepsilon, \delta, \alpha, \beta, \tau)^m, Q_{\mathrm{outer1}}(\varepsilon, \delta, \alpha, \beta, \tau)^m \Big) \quad (13)$$

*where $P_{\mathrm{outer1}}(\varepsilon, \delta, \alpha, \beta, \tau)^m$ and $Q_{\mathrm{outer1}}(\varepsilon, \delta, \alpha, \beta, \tau)^m$ are the $m$-th order product distributions of discrete random variables distributed as*

$$P_{\mathrm{outer1}}(\varepsilon, \delta, \alpha, \beta, \tau) \;=\; \left[ \tfrac{\alpha(\delta - \varepsilon) - \varepsilon\tau}{\alpha - \varepsilon}, \quad \tfrac{\alpha(\alpha + \tau - \delta)}{\alpha - \varepsilon}, \quad 1 - \tau - \alpha - \beta, \quad \beta, \quad 0 \right], \; \text{and} \quad (14)$$

$$Q_{\mathrm{outer1}}(\varepsilon, \delta, \alpha, \beta, \tau) \;=\; \left[ 0, \quad \alpha, \quad 1 - \tau - \alpha - \beta, \quad \tfrac{\beta(\beta + \tau - \delta)}{\beta - \varepsilon}, \quad \tfrac{\beta(\delta - \varepsilon) - \varepsilon\tau}{\beta - \varepsilon} \right]. \quad (15)$$

*The solution to the minimization in (3) is*

$$L_2(\tau, m) \quad \triangleq \quad \min_{\frac{\varepsilon\tau}{\delta - \varepsilon} \le \alpha \le 1 - \frac{\delta\tau}{\delta - \varepsilon}} \; d_{\mathrm{TV}}\Big( P_{\mathrm{inner}}(\alpha)^m, Q_{\mathrm{inner}}(\alpha, \tau)^m \Big), \quad (16)$$

*where $P_{\mathrm{inner}}(\alpha)$ and $Q_{\mathrm{inner}}(\alpha, \tau)$ are defined as in Theorem 4.*

*If $\delta + \varepsilon > 1$ and $\delta - \varepsilon \le \tau \le (\delta - \varepsilon)/(2 - \delta - \varepsilon)$ then the solution to the maximization in (3) is*

$$U_2(\epsilon, \delta, \tau, m) \quad \triangleq \quad \max_{\alpha + \beta \le 1 - \tau, \frac{(1 - \delta)\tau}{\delta - \varepsilon} \le \alpha, \beta} \; d_{\mathrm{TV}}\Big( P_{\mathrm{outer2}}(\varepsilon, \delta, \alpha, \beta, \tau)^m, Q_{\mathrm{outer2}}(\varepsilon, \delta, \alpha, \beta, \tau)^m \Big) \quad (17)$$

*where $P_{\mathrm{outer2}}(\varepsilon, \delta, \alpha, \beta, \tau)^m$ and $Q_{\mathrm{outer2}}(\varepsilon, \delta, \alpha, \beta, \tau)^m$ are the $m$-th order product distributions of discrete random variables distributed as*

$$P_{\mathrm{outer2}}(\varepsilon, \delta, \alpha, \beta, \tau) \;=\; \left[ \tfrac{\alpha(\delta - \varepsilon) - (1 - \delta)\tau}{\alpha - (1 - \delta)}, \quad \tfrac{\alpha(\alpha + \tau - (1 - \varepsilon))}{\alpha - (1 - \delta)}, \quad 1 - \tau - \alpha - \beta, \quad \beta, \quad 0 \right], \; \text{and} \quad (18)$$

$$Q_{\mathrm{outer2}}(\varepsilon, \delta, \alpha, \beta, \tau) \;=\; \left[ 0, \quad \alpha, \quad 1 - \tau - \alpha - \beta, \quad \tfrac{\beta(\beta + \tau - (1 - \varepsilon))}{\beta - (1 - \delta)}, \quad \tfrac{\beta(\delta - \varepsilon) - (1 - \delta)\tau}{\beta - (1 - \delta)} \right]. \quad (19)$$

*The solution to the minimization in (3) is*

$$L_3(\tau, m) \quad \triangleq \quad \min_{\frac{(1 - \delta)\tau}{\delta - \varepsilon} \le \alpha \le 1 - \frac{(1 - \varepsilon)\tau}{\delta - \varepsilon}} \; d_{\mathrm{TV}}\Big( P_{\mathrm{inner}}(\alpha)^m, Q_{\mathrm{inner}}(\alpha, \tau)^m \Big), \quad (20)$$

*where $P_{\mathrm{inner}}(\alpha)$ and $Q_{\mathrm{inner}}(\alpha, \tau)$ are defined as in Theorem 4.*

*If $\tau > \max\{(\delta - \varepsilon)/(\delta + \varepsilon), (\delta - \varepsilon)/(2 - \delta - \varepsilon)\}$, then the optimization in (3) has no solution and the feasible set is an empty set.*

A proof of this theorem is provided in Section F.3, which also critically relies on the proposed mode collapse region representation of the pair $(P, Q)$ and the celebrated result by Blackwell from [4]. The solutions in Theorem 6 can be numerically evaluated for any given choices of $(\varepsilon, \delta, \tau)$ as we show in Figure 10.

### D.5 The benefit of packing degree $m$

We give a practitioner the choice of the degree $m$ of packing, namely how many samples to jointly pack together. There is a natural trade-off between computational complexity (which increases gracefully with $m$) and the additional distinguishability, which we illustrate via an example. Consider the goal of differentiating two families of target-generator pairs, one with mode collapse and one without:

$$H_0(\varepsilon, \delta, \tau) \triangleq \{(P, Q)|(P, Q) \text{ without } (\varepsilon, \delta)\text{-mode collapse or augmentation, and } d_{\mathrm{TV}}(P, Q) = \tau\},$$

$$H_1(\varepsilon, \delta, \tau) \triangleq \{(P, Q)|(P, Q) \text{ with } (\varepsilon, \delta)\text{-mode collapse and } d_{\mathrm{TV}}(P, Q) = \tau\}. \quad (21)$$

As both families have the same total variation distances, they cannot be distinguished by an unpacked discriminator. However, a packed discriminator that uses $m$ samples jointly can differentiate those

No $(\epsilon, \delta)$ mode collapse or augmentation

$(\epsilon, \delta) = (0.03, 0.1)$    $(\epsilon, \delta) = (0.04, 0.1)$    $(\epsilon, \delta) = (0.05, 0.1)$

$(\epsilon, \delta) = (0.06, 0.1)$    $(\epsilon, \delta) = (0.07, 0.1)$    $(\epsilon, \delta) = (0.08, 0.1)$

Figure 10: The evolution of total variation distance over the packing degree $m$ for pairs with no mode collapse/augmentation is shown as a blue band, as defined by the optimization (3) and computed using Theorem 6. For a fixed $d_{\mathrm{TV}}(P, Q) = \tau = 0.11$ and the lack of $(\varepsilon, \delta = 0.1)$-mode collapse/augmentation constraints, we show the evolution with different choices of $\varepsilon \in \{0.03, 0.04, 0.05, 0.06, 0.07, 0.08\}$. The black solid lines show the maximum/minimum total variation in the optimization (4) as a reference. The family of pairs $(P, Q)$ with weaker mode collapse (i.e. larger $\varepsilon$ in the constraint), occupies a smaller region at the bottom with smaller total variation under packing, and hence is less penalized when training the generator.

two classes and even separate them entirely for a certain choices of parameters, as illustrated in Figure 11. In red, we show the achievable $d_{\mathrm{TV}}(P^m, Q^m)$ for $H_1(\varepsilon = 0.02, \delta = 0.1, \tau = 0.11)$ (the bounds in Theorem (2)). In blue is shown a similar region for $H_0(\varepsilon = 0.05, \delta = 0.1, \tau = 0.11)$ (the bounds in Theorem (6)). Although the two families are strictly separated (one with $\varepsilon = 0.02$ and another with $\varepsilon = 0.05$), a non-packed discriminator cannot differentiate those two families as the total variation is the same for both. However, as you pack mode samples, the packed discriminator becomes more powerful in differentiating the two hypothesized families. For instance, for $m \geq 5$, the total variation distance completely separates the two families.

In general, the overlap between those regions depends on the specific choice of parameters, but the overall trend is universal: packing separates generators with mode collapse from those without. Further, as the degree of packing increases, a packed discriminator increasingly penalizes generators with mode collapse and rewards generators that exhibit less mode collapse. Even if we consider complementary sets $H_0$ and $H_1$ with the same $\varepsilon$ and $\delta$ (such that the union covers the whole space of pairs of $(P, Q)$ with the same total variation distance), the least penalized pairs will be those with least mode collapse, which fall within the blue region of the bottom right panel in Figure 10. This is consistent with our empirical observations in Section 4.

## D.6 Jensen-Shannon divergence

Our theoretical analysis focused on 0-1 loss, as our current analysis technique gives exact solutions to the optimization problems (4), (2), and (3) if the metric is total variation distance. This follows from the fact that we can provide tight inner and outer regions to the family of mode collapse regions $\mathcal{R}(P, Q)$ that have the same total variation distances as $d_{\mathrm{TV}}(P, Q)$ as shown in Section F.

In practice, 0-1 loss is never used, as it is not differentiable. A popular choice of a loss function is the cross entropy loss, which gives a metric of Jensen-Shannon (JS) divergence, as shown in the beginning of Section 3. However, the same proof techniques used to show Theorems 2 and 6 give

Figure 11: Evolution of achievable total variation distances $d_{\mathrm{TV}}(P^m, Q^m)$ over packing size $m$ for two families of the target-generator pairs $H_0(0.05, 0.1, 0.11)$ and $H_1(0.02, 0.1, 0.11)$. The mode-collapsing $H_1$ is penalized significantly by the discriminator (only with $m > 1$) and the two families can be strictly separated with packing for $m > 5$.

loose bounds on JS divergence. In particular, this gap prevents us from sharply characterizing the full effect of packing degree $m$ on the JS divergence of a pair of distributions. Nonetheless, we find that empirically, packing seems to reduce mode collapse even under a cross entropy loss. It is an interesting open question to find solutions to the optimization problems (4), (2), and (3), when the metric is the (more common) Jensen-Shannon divergence.

Although our proposed analysis technique does not provide a tight analysis for JS divergence, we can still analyze a toy example similar to the one in Section D.1. Consider a toy example with a uniform target distribution $P = U([0, 1])$ over $[0, 1]$, a mode collapsing generator $Q_1 = U([0.4, 1])$, and a non mode collapsing generator $Q_2 = 0.285\, U([0, 0.77815]) + 3.479\, U([0.77815, 1])$. They are designed to have the same Jensen-Shannon divergence, i.e. $d_{\mathrm{JS}}(P, Q_1) = d_{\mathrm{JS}}(P, Q_2) = 0.1639$, but $Q_1$ exhibits an extreme mode collapse as the whole probability mass in $[0, 0.4]$ is lost, whereas $Q_2$ captures a more balanced deviation from $P$. Figure 12 shows that the mode collapsing $Q_1$ have large JS divergence (and hence penalized more) under packing, compared to non-mode-collapsed $Q_2$.

Figure 12: Jensen-Shannon divergence increases faster as a function of packing degree $m$ for a mode collapsing generator $Q_1$, compared to a non mode collapsing generator $Q_2$.

## E    Operational interpretation of mode collapse via hypothesis testing region

So far, all the definitions and theoretical results have been explained without explicitly using the *mode collapse region*. The main contribution of introducing the region definition is that it provides a

new proof technique based on the geometric properties of these two-dimensional regions. Concretely, we show that the proposed mode collapse region is equivalent to a similar notion in binary hypothesis testing. This allows us to bring powerful mathematical tools from this mature area in statistics and information theory—in particular, the *data processing inequalities* originating from the seminal work of Blackwell [4]. We make this connection precise, which gives insights on how to interpret the mode collapse region, and list the properties and techniques which dramatically simplify the proof, while providing the tight results in Section F.

### E.1 Equivalence between the mode collapse region and the ROC curve

There is a simple one-to-one correspondence between mode collapse region as we define it in Definition 1 (e.g. Figure 2) and the ROC curve studied in binary hypothesis testing. In the classical testing context, there are two hypotheses, $h = 0$ or $h = 1$, and we make observations via some stochastic experiment in which our observations depend on the hypothesis. Let $X$ denote this observation. One way to visualize such an experiment is using a two-dimensional region defined by the corresponding type I and type II errors. Concretely, an ROC curve of a binary hypothesis testing is obtained by plotting the largest achievable true positive rate (TPR), i.e. $1-$probability of missed detection, or equivalently $1-$ type II error, on the vertical axis against the false positive rate (FPR), i.e probability of false alarm or equivalently type I error, on the horizontal axis.

We can map this binary hypothesis testing setup directly to the GAN context. Suppose the null hypothesis $h = 0$ denotes observations being drawn from the generated distribution $Q$, and the alternate hypothesis $h = 1$ denotes observations being drawn from the true distribution $P$. Given a sample $X$ from this experiment, suppose we make a decision on whether the sample came from $P$ or $Q$ based on a rejection region $S_{\text{reject}}$, such that we reject the null hypothesis if $X \in S_{\text{reject}}$. FPR (i.e. Type I error) is when the null hypothesis is true but rejected, which happens with $\mathbb{P}(X \in S_{\text{reject}}|h = 0)$, and TPR (i.e. 1-type II error) is when the null hypothesis is false but accepted, which happens with $\mathbb{P}(X \in S_{\text{reject}}|h = 1)$. Sweeping through the achievable pairs $(\mathbb{P}(X \in S_{\text{reject}}|h = 1), \mathbb{P}(X \in S_{\text{reject}}|h = 0))$ for all rejection sets, this defines a two-dimensional convex region that we call *hypothesis testing region*. The upper boundary of this convex set is the ROC curve. An example of ROC curves for the two toy examples $(P, Q_1)$ and $(P, Q_2)$ from Figure 2 are shown in Figure 13.

Figure 13: The hypothesis testing region of $(P, Q)$ (bottom row) is the same as the mode collapse region (top row). We omit the region above $y = x$ axis in the hypothesis testing region as it is symmetric. The regions for mode collapsing toy example in Figure 2 $(P, Q_1)$ are shown on the left and the regions for the non mode collapsing example $(P, Q_2)$ are shown on the right.

In defining the region, we allow stochastic decisions, such that if a point $(x, y)$ and another point $(x', y')$ are achievable TPR and FPR, then any convex combination of those points are also achievable by randomly choosing between those two rejection sets. Hence, the resulting hypothesis testing region is always a convex set by definition. We also show only the region below the 45-degree line passing through $(0,0)$ and $(1,1)$, as the other region is symmetric and redundant. For a given pair $(P, Q)$, there is a very simple relation between its mode collapse region and hypothesis testing region.

**Remark 7** (Equivalence). *For a pair of target $P$ and generator $Q$, the hypothesis testing region is the same as the mode collapse region.*

This follows immediately from the definition of mode collapse region in Definition 1. If there exists a set $S$ such that $P(S) \geq \delta$ and $Q(S) \leq \varepsilon$, then for the choice of $S_{\text{reject}} = S$ in the binary hypothesis testing, there the point $(\mathbb{P}(X \in S_{\text{reject}}|h = 0) = \varepsilon, \mathbb{P}(X \in S_{\text{reject}}|h = 1) = \delta)$ in the hypothesis testing region is achievable. The converse is also true, in the case we make deterministic decisions on $S_{\text{reject}}$. As the mode collapse region is defined as a convex hull of all achievable points, the points in the hypothesis testing region that require randomized decisions can also be covered.

For example, the hypothesis testing regions of the toy examples from Figure 2 are shown below in Figure 13. This simple relation allows us to tap into the rich analysis tools known for hypothesis testing regions and ROC curves. We list such properties of mode collapse regions derived from this relation in the next section. The proof of all the remarks follow from the equivalence to binary hypothesis testing and corresponding existing results from [4] and [12].

### E.2  Properties of the mode collapse region

Given the equivalence between the mode collapse region and the binary hypothesis testing region, several important properties follow as corollaries. First, the hypothesis testing region is a sufficient statistic for the purpose of binary hypothesis testing from a pair of distributions $(P, Q)$. This implies, among other things, that all $f$-divergences can be derived from the region. In particular, for the purpose of GAN with 0-1 loss, we can define total variation as a geometric property of the region, which is crucial to proving our main results.

**Remark 8** (Total variation distance). *The total variation distance between $P$ and $Q$ is the intersection between the vertical axis and the tangent line to the upper boundary of $\mathcal{R}(P, Q)$ that has a slope of one, as shown in Figure 14.*

Figure 14: Total variation distance is one among many properties of $(P, Q_2)$ that can be directly read off of the region $\mathcal{R}(P, Q)$.

This follows from the equivalence of the mode collapse region (Remark 7) and the hypothesis testing region. This geometric definition of total variation allows us to enumerate over all pairs $(P, Q)$ that have the same total variation $\tau$ in our proof, via enumerating over all regions that touch the line that has a unit slope and a shift $\tau$ (see Figure 15).

The major strength of the region perspective, as originally studied by Blackwell [4], is in providing a comparison of stochastic experiments. In our GAN context, consider comparing two pairs of target distributions and generators $(P, Q)$ and $(P', Q')$ as follows. First, a hypothesis $h$ is drawn, choosing whether to produce samples from the true distribution, in which case we say $h = 1$, or to produce samples from the generator, in which case we say $h = 0$. Conditioned on this hypothesis $h$, we use $X$ to denote a random variable that is drawn from the first pair $(P, Q)$ such that $f_{X|h}(x|1) = P(x)$

and $f_{X|h}(x|0) = Q(x)$. Similarly, we use $X'$ to denote a random sample from the second pair, where $f_{X'|h}(x|1) = P'(x)$ and $f_{X'|h}(x|0) = Q'(x)$. Note that the conditional distributions are well-defined for both $X$ and $X'$, but there is no coupling defined between them. Suppose $h$ is independently drawn from the uniform distribution.

**Definition 9.** *For a given coupling between $X$ and $X'$, we say $X$ dominates $X'$ if they form a Markov chain $h$–$X$–$X'$.*

The *data processing inequality* in the following remark shows that if we further *process* the output samples from the pair $(P, Q)$ then the further processed samples can only have less mode collapse. Processing output of stochastic experiments has the effect of smoothing out the distributions, and mode collapse, which corresponds to a *peak* in the pair of distributions, are smoothed out in the processing down the Markov chain.

**Remark 10** (Data processing inequality). *The following data processing inequality holds for the mode collapse region. For two coupled target-generator pairs $(P, Q)$ and $(P', Q')$, if $X$ dominates another pair $X'$, then*

$$\mathcal{R}(P', Q') \subseteq \mathcal{R}(P, Q) .$$

This is expected, and follows directly from the equivalence of the mode collapse region (Remark 7) and the hypothesis testing region, and corresponding data processing inequality of hypothesis testing region in [12]. What is perhaps surprising is that the reverse is also true.

**Remark 11** (Reverse data processing inequality). *The following reverse data processing inequality holds for the mode collapse region. For two paired marginal distributions $X$ and $X'$, if*

$$\mathcal{R}(P', Q') \subseteq \mathcal{R}(P, Q) ,$$

*then there exists a coupling of the random samples from $X$ and $X'$ such that $X$ dominates $X'$, i.e. they form a Markov chain $h$–$X$–$X'$.*

This follows from the equivalence between the mode collapse region and the hypothesis testing region (Remark 7) and Blackwell's celebrated result on comparisons of stochastic experiments [4] (see [12] for a simpler version of the statement). This region interpretation, and the accompanying (reverse) data processing inequality, abstracts away all the details about $P$ and $Q$, enabling us to use geometric analysis tools to prove our results. In proving our main results, we will mainly rely on the following remark, which is the corollary of the Remarks 10 and 11.

**Remark 12.** *For all positive integers $m$, the dominance of regions are preserved under taking $m$-th order product distributions, i.e. if $\mathcal{R}(P', Q') \subseteq \mathcal{R}(P, Q)$, then $\mathcal{R}((P')^m, (Q')^m) \subseteq \mathcal{R}(P^m, Q^m)$.*

## F  Proofs of the main results

In this section, we showcase how the region interpretation provides a new proof technique that is simple and tight. This transforms the measure-theoretic problem into a geometric one in a simple 2D compact plane, facilitating the proof of otherwise-challenging results.

### F.1  Proof of Theorem 4

Note that although the original optimization (4) has nothing to do with mode collapse, we use the mode collapse region to represent the pairs $(P, Q)$ to be optimized over. This allows us to use simple geometric techniques to enumerate over all possible pairs $(P, Q)$ that have the same total variation distance $\tau$.

By Remark 8, all pairs $(P, Q)$ that have total variation $\tau$ must have a mode collapse region $\mathcal{R}(P, Q)$ that is tangent to the blue line in Figure 15. Let us denote a point where $\mathcal{R}(P, Q)$ meets the blue line by the point $(1 - \alpha - \tau, 1 - \alpha)$ in the 2D plane, parametrized by $\alpha \in [0, 1 - \tau]$. Then, for any such $(P, Q)$, we can sandwich the region $\mathcal{R}(P, Q)$ between two regions $\mathcal{R}_{\text{inner}}$ and $\mathcal{R}_{\text{outer}}$:

$$\mathcal{R}_{\text{inner}}(\alpha, \tau) \subseteq \mathcal{R}(P, Q) \subseteq \mathcal{R}_{\text{outer}}(\tau) , \tag{22}$$

which are illustrated in Figure 16. Now, we wish to understand how these inner and outer regions evolve under product distributions. This endeavor is complicated by the fact that there can be

Figure 15: For any pair $(P, Q)$ with total variation distance $\tau$, there exists an $\alpha$ such that the corresponding region $\mathcal{R}(P, Q)$ is sandwiched between $\mathcal{R}_{\text{inner}}(\alpha, \tau)$ and $\mathcal{R}_{\text{outer}}(\tau)$.

infinite pairs of distributions that have the same region $\mathcal{R}(P, Q)$. However, note that if two pairs of distributions have the same region $\mathcal{R}(P, Q) = \mathcal{R}(P', Q')$, then their product distributions will also have the same region $\mathcal{R}(P^m, Q^m) = \mathcal{R}((P')^m, (Q')^m)$. As such, we can focus on the simplest, *canonical* pair of distributions, whose support set has the minimum cardinality over all pairs of distributions with region $\mathcal{R}(P, Q)$.

For a given $\alpha$, we denote the pairs of canonical distributions achieving these exact inner and outer regions as in Figure 16: let $(P_{\text{inner}}(\alpha), Q_{\text{inner}}(\alpha, \tau))$ be as defined in (6) and (7), and let $(P_{\text{outer}}(\tau), Q_{\text{outer}}(\tau))$ be defined as below. Since the outer region has three sides (except for the universal 45-degree line), we only need alphabet size of three to find the canonical probability distributions corresponding to the outer region. By the same reasoning, the inner region requires only a binary alphabet. Precise probability mass functions on these discrete alphabets can be found easily from the shape of the regions and the equivalence to the hypothesis testing region explained in Section E.

Figure 16: Canonical pairs of distributions corresponding to $\mathcal{R}_{\text{inner}}(\alpha, \tau)$ and $\mathcal{R}_{\text{outer}}(\tau)$.

By the preservation of dominance under product distributions in Remark 12, it follows from the dominance in (22) that for any $(P, Q)$ there exists an $\alpha$ such that

$$\mathcal{R}(P_{\text{inner}}(\alpha)^m, Q_{\text{inner}}(\alpha, \tau)^m) \ \subseteq \ \mathcal{R}(P^m, Q^m) \ \subseteq \ \mathcal{R}(P_{\text{outer}}(\tau)^m, Q_{\text{outer}}(\tau)^m) . \quad (23)$$

Due to the data processing inequality of mode collapse region in Remark 11, it follows that dominance of region implies dominance of total variation distances:

$$\min_{0 \leq \alpha \leq 1-\tau} d_{\text{TV}}(P_{\text{inner}}(\alpha)^m, Q_{\text{inner}}(\alpha, \tau)^m) \ \leq \ d_{\text{TV}}(P^m, Q^m) \ \leq \ d_{\text{TV}}(P_{\text{outer}}(\tau)^m, Q_{\text{outer}}(\tau)^m) . \quad (24)$$

The RHS and LHS of the above inequalities can be completely characterized by taking the $m$-th power of those canonical pairs of distributions. For the upper bound, all mass except for $(1 - \tau)^m$ is nonzero only on one of the pairs, which gives $d_{\text{TV}}(P_{\text{outer}}^m, Q_{\text{outer}}^m) = 1 - (1 - \tau)^m$. For the lower bound, writing out the total variation gives $L(\tau, m)$ in (5). This finishes the proof of Theorem 4.

### F.2   Proof of Theorem 2

In optimization (2), we consider only those pairs with $(\varepsilon, \delta)$-mode collapse. It is simple to see that the outer bound does not change. We only need a new inner bound. Let us denote a point where $\mathcal{R}(P,Q)$ meets the blue line by the point $(1 - \alpha - \tau, 1 - \alpha)$ in the 2D plane, parametrized by $\alpha \in [0, 1 - \tau]$. We consider the case where $\alpha < 1 - (\tau\delta/(\delta - \varepsilon))$ for now, and treat the case when $\alpha$ is larger separately, as the analyses are similar but require a different canonical pair of distributions $(P, Q)$ for the inner bound. The additional constraint that $(P, Q)$ has $(\varepsilon, \delta)$-mode collapse translates into a geometric constraint that we need to consider all regions $\mathcal{R}(P, Q)$ that include the orange solid circle at point $(\varepsilon, \delta)$. Then, for any such $(P, Q)$, we can sandwich the region $\mathcal{R}(P, Q)$ between two regions $\mathcal{R}_{\text{inner1}}$ and $\mathcal{R}_{\text{outer}}$:

$$\mathcal{R}_{\text{inner1}}(\varepsilon, \delta, \alpha, \tau) \ \subseteq \ \mathcal{R}(P, Q) \ \subseteq \ \mathcal{R}_{\text{outer}}(\tau) \,, \tag{25}$$

Figure 17: For any pair $(P, Q)$ with $(\varepsilon, \delta)$-mode collapse, the corresponding region $\mathcal{R}(P, Q)$ is sandwiched between $\mathcal{R}_{\text{inner1}}(\varepsilon, \delta, \alpha, \tau)$ and $\mathcal{R}_{\text{outer}}(\tau)$.

Let $(P_{\text{inner1}}(\delta, \alpha), Q_{\text{inner1}}(\varepsilon, \alpha, \tau))$ defined in (9) and (10), and $(P_{\text{outer}}(\tau), Q_{\text{outer}}(\tau))$ defined in Section F.1 denote the pairs of canonical distributions achieving the inner and outer regions exactly as shown in Figure 18. By the preservation of dominance under product distributions in Remark 12,

Figure 18: Canonical pairs of distributions corresponding to $\mathcal{R}_{\text{inner}}(\varepsilon, \delta, \tau, \alpha)$ and $\mathcal{R}_{\text{outer}}(\tau)$.

it follows from the dominance in (25) that for any $(P, Q)$ there exists an $\alpha$ such that

$$\mathcal{R}(P_{\text{inner1}}(\delta, \alpha)^m, Q_{\text{inner1}}(\varepsilon, \delta, \alpha, \tau)^m) \ \subseteq \ \mathcal{R}(P^m, Q^m) \ \subseteq \ \mathcal{R}(P_{\text{outer}}(\tau)^m, Q_{\text{outer}}(\tau)^m) \,. \tag{26}$$

Due to the data processing inequality of mode collapse region in Remark 11, it follows that dominance of region implies dominance of total variation distances:

$$\min_{0 \le \alpha \le 1 - \frac{\tau\delta}{\delta - \varepsilon}} d_{\text{TV}}(P_{\text{inner1}}(\delta, \alpha)^m, Q_{\text{inner1}}(\varepsilon, \delta, \alpha, \tau)^m) \le \ d_{\text{TV}}(P^m, Q^m) \ \le \ d_{\text{TV}}(P_{\text{outer}}(\tau)^m, Q_{\text{outer}}(\tau)^m) \,.$$
$$\tag{27}$$

The RHS and LHS of the above inequalities can be completely characterized by taking the $m$-th power of those canonical pairs of distributions. For the upper bound, all mass except for $(1 - \tau)^m$ is

nonzero only on one of the pairs, which gives $d_{\mathrm{TV}}(P_{\mathrm{outer}}^m, Q_{\mathrm{outer}}^m) = 1 - (1 - \tau)^m$. For the lower bound, writing out the total variation gives $L_1(\varepsilon, \delta, \tau, m)$ in (8).

For $\alpha > 1 - (\tau\delta/(\delta - \varepsilon))$, we need to consider a different class of canonical distributions for the inner region, shown below. The inner region $\mathcal{R}_{\mathrm{inner2}}(\alpha, \tau)$ and corresponding canonical distributions $P_{\mathrm{inner2}}(\alpha)$ and $Q_{\mathrm{inner2}}(\alpha, \tau)$ defined in (11) and (12) are shown below. We take the smaller one between the total variation distance resulting from these two cases. Note that $\alpha \leq 1 - \tau$ by definition. This finishes the proof of Theorem 2.

Figure 19: When $\alpha > 1 - (\tau\delta/(\delta - \varepsilon))$, this shows a canonical pair of distributions corresponding to $\mathcal{R}_{\mathrm{inner}}(\varepsilon, \delta, \tau, \alpha)$ for the mode-collapsing scenario $H_1(\varepsilon, \delta, \tau)$.

### F.3 Proof of Theorem 6

When $\tau < \delta - \varepsilon$, all pairs $(P, Q)$ with $d_{\mathrm{TV}}(P, Q) = \tau$ cannot have $(\varepsilon, \delta)$-mode collapse, and the optimization of (3) reduces to that of (4) without any mode collapse constraints.

When $\delta + \varepsilon \leq 1$ and $\tau > (\delta - \varepsilon)/(\delta + \varepsilon)$, no convex region $\mathcal{R}(P, Q)$ can touch the 45-degree line at $\tau$ as shown below, and the feasible set is empty. This follows from the fact that a triangle region passing through both $(\varepsilon, \delta)$ and $(1 - \delta, 1 - \varepsilon)$ will have a total variation distance of $(\delta - \varepsilon)/(\delta + \varepsilon)$. Note that no $(\varepsilon, \delta)$-mode augmentation constraint translates into the region not including the point $(1 - \delta, 1 - \varepsilon)$. We can see easily from Figure 20 that any total variation beyond that will require violating either the no-mode-collapse constraint or the no-mode-augmentation constraint. Similarly, when $\delta + \varepsilon > 1$ and $\tau > (\delta - \varepsilon)/(2 - \delta - \varepsilon)$, the feasible set is also empty. These two can be unified as $\tau > \max\{(\delta - \varepsilon)/(\delta + \varepsilon), (\delta - \varepsilon)/(2 - \delta - \varepsilon)\}$.

Figure 20: When $\delta + \varepsilon \leq 1$ and $\tau = (\delta - \varepsilon)/(\delta + \varepsilon)$ (i.e. $(1 - \tau)/2 : (1 + \tau)/2 = \varepsilon : \delta$), a triangle mode collapse region that touches both points $(\varepsilon, \delta)$ and $(1 - \delta, 1 - \varepsilon)$ at two of its edges also touches the 45-degree line with a $\tau$ shift at a vertex (left). When $\delta + \varepsilon > 1$, the same happens when $\tau = (\delta - \varepsilon)/(2 - \delta - \varepsilon)$ (i.e. $(1 - \tau)/2 : (1 + \tau)/2 = (1 - \delta) : (1 - \varepsilon)$). Hence, if $\tau > \max\{(\delta - \varepsilon)/(\delta + \varepsilon), (\delta - \varepsilon)/(2 - \delta - \varepsilon)\}$, then the triangle region that does not include both orange points cannot touch the blue 45-degree line.

Suppose $\delta + \varepsilon \leq 1$, and consider the intermediate regime when $\delta - \varepsilon \leq \tau \leq (\delta - \varepsilon)/(\delta + \varepsilon)$. In optimization (3), we consider only those pairs with no $(\varepsilon, \delta)$-mode collapse or $(\varepsilon, \delta)$-mode

augmentation. It is simple to see that the inner bound does not change from optimization in (4). Let us denote a point where $\mathcal{R}(P,Q)$ meets the blue line by the point $(1-\alpha'-\tau, 1-\alpha')$ in the 2D plane, parametrized by $\alpha' \in [0, 1-\tau]$. The $\mathcal{R}(\alpha', \tau)$ defined in Figure 16 works in this case also. We only need a new outer bound.

We construct an outer bound region, according to the following rule. We fit a hexagon where one edge is the 45-degree line passing through the origin, one edge is the vertical axis, one edge is the horizontal line passing through $(1,1)$, one edge is the 45-degree line with shift $\tau$ shown in blue in Figure 21, and the remaining two edges include the two orange points, respectively, at $(\varepsilon, \delta)$ and $(1-\delta, 1-\varepsilon)$. For any $\mathcal{R}(P,Q)$ satisfying the constraints in (3), there exists at least one such hexagon that includes $\mathcal{R}(P,Q)$. We parametrize the hexagon by $\alpha$ and $\beta$, where $(\alpha, \tau+\alpha)$ denotes the left-most point where the hexagon meets the blue line, and $(1-\tau-\beta, 1-\beta)$ denotes the right-most point where the hexagon meets the blue line.

The additional constraint that $(P,Q)$ has no $(\varepsilon, \delta)$-mode collapse or $(\varepsilon, \delta)$-mode augmentation translates into a geometric constraint that we need to consider all regions $\mathcal{R}(P,Q)$ that does not include the orange solid circle at point $(\varepsilon, \delta)$ and $(1-\delta, 1-\varepsilon)$. Then, for any such $(P,Q)$, we can sandwich the region $\mathcal{R}(P,Q)$ between two regions $\mathcal{R}_{\text{inner}}$ and $\mathcal{R}_{\text{outer1}}$:

$$\mathcal{R}_{\text{inner}}(\alpha', \tau) \subseteq \mathcal{R}(P,Q) \subseteq \mathcal{R}_{\text{outrer1}}(\varepsilon, \delta, \alpha, \beta, \tau), \tag{28}$$

where $\mathcal{R}_{\text{inner}}(\alpha, \tau)$ is defined as in Figure 16.

Figure 21: For any pair $(P,Q)$ with no $(\varepsilon, \delta)$-mode collapse or no $(\varepsilon, \delta)$-mode augmentation, the corresponding region $\mathcal{R}(P,Q)$ is sandwiched between $\mathcal{R}_{\text{inner}}(\alpha', \tau)$ and $\mathcal{R}_{\text{outer1}}(\varepsilon, \delta, \alpha, \beta, \tau)$ (left). A canonical pair of distributions corresponding to $\mathcal{R}_{\text{outer1}}(\varepsilon, \delta, \alpha, \beta, \tau)$ (middle and right).

Let $(P_{\text{inner}}(\alpha'), Q_{\text{inner}}(\alpha', \tau))$ defined in (6) and (7), and $(P_{\text{outer1}}(\varepsilon, \delta, \alpha, \beta, \tau), Q_{\text{outer1}}(\varepsilon, \delta, \alpha, \beta, \tau))$ denote the pairs of canonical distributions achieving the inner and outer regions exactly as shown in Figure 21.

By the preservation of dominance under product distributions in Remark 12, it follows from the dominance in (28) that for any $(P,Q)$ there exist $\alpha'$, $\alpha$, and $\beta$ such that

$$\mathcal{R}(P_{\text{inner}}(\alpha')^m, Q_{\text{inner}}(\alpha', \tau)^m) \subseteq \mathcal{R}(P^m, Q^m) \subseteq \mathcal{R}(P_{\text{outer1}}(\varepsilon, \delta, \alpha, \beta, \tau)^m, Q_{\text{outer1}}(\varepsilon, \delta, \alpha, \beta, \tau)^m). \tag{29}$$

Due to the data processing inequality of mode collapse region in Remark 11, it follows that dominance of region implies dominance of total variation distances:

$$\min_{\frac{\varepsilon\tau}{\delta-\varepsilon} \le \alpha' \le 1 - \frac{\tau\delta}{\delta-\varepsilon}} d_{\text{TV}}(P_{\text{inner}}(\alpha')^m, Q_{\text{inner}}(\alpha', \tau)^m) \le d_{\text{TV}}(P^m, Q^m)$$

$$\le \max_{\alpha, \beta \ge \frac{\varepsilon\tau}{\delta-\varepsilon}, \alpha+\beta \le 1-\tau} d_{\text{TV}}(P_{\text{outer1}}(\varepsilon, \delta, \alpha, \beta, \tau)^m, Q_{\text{outer1}}(\varepsilon, \delta, \alpha, \beta, \tau)^m). \tag{30}$$

The RHS and LHS of the above inequalities can be completely characterized by taking the $m$-th power of those canonical pairs of distributions, and then taking the respective minimum over $\alpha'$ and maximum over $\alpha$ and $\beta$. For the upper bound, this gives $U_1(\epsilon, \delta, \tau, m)$ in (13), and for the lower bound this gives $L_2(\tau, m)$ in (16).

Now, suppose $\delta + \varepsilon > 1$, and consider the intermediate regime when $\delta - \varepsilon \le \tau \le (\delta - \varepsilon)/(2 - \delta - \varepsilon)$. We have a different outer bound $\mathcal{R}_{\text{outer2}}(\varepsilon, \delta, \alpha, \delta, \tau)$ as the role of $(\varepsilon, \delta)$ and $(1 - \delta, 1 - \varepsilon)$ have switched. A similar analysis gives

$$d_{\text{TV}}(P^m, Q^m) \ \le \ \max_{\alpha, \beta \ge \frac{(1-\delta)\tau}{\delta-\varepsilon}, \alpha+\beta \le 1-\tau} \ d_{\text{TV}}(P_{\text{outer2}}(\varepsilon, \delta, \alpha, \beta, \tau)^m, Q_{\text{outer2}}(\varepsilon, \delta, \alpha, \beta, \tau)^m) \ , \tag{31}$$

where the canonical distributions are shown in Figure 22 and defined in (18) and (19). This gives $U_2(\epsilon, \delta, \tau, m)$ in (17). For the lower bound we only need to change the range of $\alpha$ we minimize over, which gives $L_3(\tau, m)$ in (20).

Figure 22: A canonical pair of distributions corresponding to $\mathcal{R}_{\text{outer2}}(\varepsilon, \delta, \alpha, \beta, \tau)$.

## Footnotes

[2] https://github.com/IshmaelBelghazi/ALI

[3] https://github.com/openai/improved-gan