[Reviews · NeurIPS 2018]

Reviewer 1



Summary : While Generative Adversarial Networks (GANs) have become the desired choice for generative tasks in the community, they also suffer from a nagging issue of mode collapse (cf. [1, 2, 3, 4] in the paper). The current literature also has some empirical ways to handle this issue (cf. [1-3, 16, 19, 22] in the paper). The paper showcases a significant stride in presenting for the first time a theoretical consolidation and treatment of the mode collapse problem. They present the technique of packing, in which the discriminator now uses multiple samples in its task. They also validate their formulation on synthetic datasets and real datasets (Stacked MNIST and CelebA). Detailed Comments: Clarity : The paper is very well written, both rigor and intuitive expositions are presented. Originality : As explained above in summary, perhaps this is the first time a framework of mode collapse is constructed and its theoretical underpinnings are discussed. While packing is in a similar vein to the already known, minibatch discrimination (as also agreed by author), their unique standing point is the easier implementation and their theoretical analysis. Quality and Significance : The paper presents the exposition very systematically and rigorously, as they mention in the introduction, their significance is conceptual, analytical and algorithmic as well. The survey of the related work is also quite comprehensive and comparative, boosting the quality of the presentation. The theoretical analysis is tied to how the total variation distance would change on packing. The authors argue (rigorously) that with packing, the total variation distance for mode collapsed distributions is amplified and hence penalized in the GAN training - thus guaranteeing resolution of the mode collapse issue. This they corroborate with experiments. I think this is a strong contribution to NIPS this year. Paper should be accepted. Some of the concerns mention below must be addressed, though they don't point to the paper's rejection: (a) Insufficient Comparison with Minibatch Discrimination : The authors do mention their advantage over the related technique of minimatch discrimination, however this is treated quite sparsely. Readers must be interested in knowing the comparison in sufficient detail especially in their implementations which is claimed to be superior for PacGAN on Page 3. (b) JS Divergence : The authors mention (in supplementary material) that their methodology breaks for JS Divergence giving loose bounds. Nonetheless given the importance of JS Divergence (which is obtained after using cross-entropy loss, which is actually how GAN is trained), it will be worthwhile to present how these bounds look like. In short, somewhat more treatment (heuristic or rigorous) to bridge the analysis between 0-1 and cross entropy loss is desired. (c) More Experimental Evidence : Given the importance and the significance of the mode collapse problem, it would be advisable to run the algorithm on several more real datasets to pronounce its robustness.

Reviewer 2



This paper considers the idea of using a mini-batch of samples to train the discriminator (packing) to mitigate the mode collapse issue in training GANs. This idea has been around in the literature and is known as mini-batch discrimination [1]. As the authors have mentioned, there are many ways for applying this technique. The authors propose a very simple implementation that just uses a fully connected layer in the input to connect the packed samples to the discriminator network. It is surprising and interesting that such a simple implementation works in practice. It seems to be a counter-intuitive structure as it does not preserve the locality of the pixel information in samples of a batch. I do not know if this would be problematic in larger experiments that make use of convolutional units at the input layers of their discriminator. Moreover, such a structure is not necessarily permutation invariant, i.e. does not treat all the samples in the batch symmetrically. Analysis: The paper seems to be exaggerating the importance and implications of the theoretical analysis. First of all the analysis only applies to the TV distance. Note that TV distance is not used in practice for training GANs as it is not practical. Moreover, the kind of relationship that is proved between mode collapse and packing in the analysis seems to be objective dependent and might not be true for some plausible objectives. A simple example to see that such a relationship does not hold is when some variant of KL divergence is used as an objective. Note that KL(P^N, Q^N) = N KL(P,Q). Therefore, packing in this case does not seem to have any effect in changing the geometry of the objective. Thus, it is not appropriate to make general claims about the fundamental relationship between packing and mode collapse (see lines 10-11) based on these limited theoretical results. The statements of the theorems 3 and 4 are very difficult to digest. Unfortunately, no intuition or clarification is provided to make them more understandable. Moreover, from the results of theorem 2 and 3, it is not readily clear that for a given epsilon, delta and tau if the lower bound of Theorem 2 would be greater than the upper-bound in Theorem 3 for some packing order m. Note that this is analytically crucial in order to make sure that a packed discriminator can distinguish mode collapse. This point does not seem to be proven explicitly, but has been mentioned as an implication of theorem 2 and 3; see lines 206-210 and 218-221. It seems that the authors only justify this point using plots in Figure 3 and Figure 11. Note that the epsilon is different for red and blue curves in Figure 3 and 11. Experiments: In the experimental section, the authors have done a good job of isolating the effects of packing on mode collapse. The improvements in mode collapse reduction, compared to the baseline methods, seem to be substantial, especially in the stacked MNIST and celebA experiments. However, the set of baseline methods do not include the most stable/state of the art methods/objectives for GANs. For example, the more stable set of methods that use the Wasserstein family of distances with some regularization, e.g. [2], are absent from the experiments. Therefore, it is difficult to confirm that such a large improvement is not due to the instability of the baseline methods in training. Moreover, no comparison with the prior methods that use mini-batch discrimination is presented. This comparison is crucial because those methods stem from the same idea as packing. It is also important to include some experiments on larger/more challenging GAN datasets such as CIFAR-10 or ImageNet. Note that the current set of experiments are more focused on evaluating the mode collapse. But the quality/sharpness of the generated samples is also important in generative models. For these larger datasets, the metrics such as inception or FID score could be used to measure the quality of the produced images as well as the mode collapse problems. The authors also claim that the introduction of packing has little computational overhead. It is not readily obvious that the introduction of packing does not affect the convergence speed of the methods. It would be very useful if some experimental results are presented to prove the above claim. Presentation: In terms of clarity and presentation, the paper requires a lot of improvement. The authors keep mentioning the relationship between the ROC curve and mode collapse region as a fundamental and intuitive point in understanding the analysis. If that is really the case, maybe it would be better to reshuffle the content so that such this point is presented in the main body. As it was mentioned above, the theorems are not clearly presented and the reader is left alone to make sense of very sophisticated expressions without any clarifications or intuitions. Some minor comments: beta is not defined in Theorem 3. - In Figure 11, even with different epsilons, at least a packing of order 5 is required to distinguish mode collapse. It would be interesting if the authors can provide a relationship between the required order of packing and the epsilon gap in H1 and H0. [1] Salimans, Tim, et al. "Improved techniques for training GANs." Advances in Neural Information Processing Systems. 2016. [2] Gulrajani, Ishaan, et al. "Improved training of Wasserstein GANs." Advances in Neural Information Processing Systems. 2017.

Reviewer 3



post-rebuttal: I feel the paper does sufficient work to warrant an accept. The main concern is regarding how well the idea of packing works with advanced and stable versions of GAN and also with other inference methods and other generative models. If the authors can alleviate this concern in the next version of the paper that would be great. Summary: This paper proposes to tackle the problem of mode collapse in Generative Adversarial Networks GAN. The problem of mode collapse is the one when the probability distribution learned by an inference algorithm (in this case GAN) is single-mode as opposed to the true probability distribution that multi-mode. This leads to low-diversity sampling of points from the learned generative model (GAN in this case). The paper proposes a simple solution to this for GAN which is force the discriminator to judge/make decision on multiple samples instead of one as in traditional GAN setting. By forcing the generator to proceed multiple high quality samples the paper claims it reduces mode collapse. The paper also provided a good formalization of the mode collapse problem and an empirical evaluation that shows that their approach leads to relaxes the mode collapse problems in GANs. Quality: I think this is a high quality paper with a nice new idea and solid formulation of the problem. Clarity: The paper overall was clear to me except the figure. I struggled to understand Figure 3, a little more explanation would be useful. Originality: I believe the paper presents original idea, algorithm, formalization and experimental results. I am quite happy with the originality of this paper. Significance: The paper tackles an important problem to the ML community and presents a nice simple solution to it. Strength: I think the main strength of the paper is the nice simple idea of packing, that is, to use multiple samples instead of one sample in a GAN setting. I am also impressed by the formalized definition of mode collapse. Weakness: I think the paper can be improved by comparing/extending its solution to generative models other than GANs. The main question that the paper does not tackle is that the problem of mode collapse is it true for other generative models as well? For which generative model is it less severe. How the PacGAN perform in comparison to those models. Other minor issues: Some parts of the paper are vague, for example, paper keeps referring to results in a vague manner as … provides novel geometric analysis technique to solve non-convex optimization problems … I did not get this claim at all (how?) Related work could be pushed to end of the paper (makes it easy to refer to what paper already presented instead of what it will present) Overall, I am quite happy with this paper.